# Towards a "universal translator" for neural dynamics at single-cell, single-spike resolution

**Yizi Zhang**[1]   **Yanchen Wang**[1]   **Donato M. Jiménez-Benetó**[2]   **Zixuan Wang**[1,3]
**Mehdi Azabou**[4]   **Blake Richards**[5]   **Renee Tung**[1]   **Olivier Winter**[6]
**The International Brain Laboratory**[7]   **Eva Dyer**[4]   **Liam Paninski**[1]   **Cole Hurwitz**[1]

[1]Columbia University   [2]Universitat Politècnica de Catalunya
[3]Zhejiang University   [4]Georgia Institute of Technology   [5]Mila, McGill University
[6]Champalimaud Foundation   [7]The International Brain Laboratory

## Abstract

Neuroscience research has made immense progress over the last decade, but our understanding of the brain remains fragmented and piecemeal: the dream of probing an arbitrary brain region and automatically reading out the information encoded in its neural activity remains out of reach. In this work, we build towards a first foundation model for neural spiking data that can solve a diverse set of tasks across multiple brain areas. We introduce a novel self-supervised modeling approach for population activity in which the model alternates between masking out and reconstructing neural activity across different time steps, neurons, and brain regions. To evaluate our approach, we design unsupervised and supervised prediction tasks using the International Brain Laboratory repeated site dataset, which is comprised of Neuropixels recordings targeting the same brain locations across 48 animals and experimental sessions. The prediction tasks include single-neuron and region-level activity prediction, forward prediction, and behavior decoding. We demonstrate that our multi-task-masking (MtM) approach significantly improves the performance of current state-of-the-art population models and enables multi-task learning. We also show that by training on multiple animals, we can improve the generalization ability of the model to unseen animals, paving the way for a foundation model of the brain at single-cell, single-spike resolution. Project page and code: https://ibl-mtm.github.io/

## 1   Introduction

Recent studies in experimental neuroscience suggest that neural computation is highly distributed across the brain and sparse in nature [34, 23]. Much of our current understanding of the brain, however, originates from studying small circuits of neurons in hand-selected brain areas during stereotyped behaviors. This has resulted in the development of neural population models that are often brain region-specific and narrowly crafted for particular experimental contexts, limiting their broader applicability and insights into distributed brain function [38]. The arrival of multi-animal neural datasets that span hundreds of interconnected brain regions [18, 17] necessitates the development of a more general approach for building neural population models.

To address these challenges, recent efforts have been directed towards building models that can be trained on neural data collected across multiple sessions and animals [3, 42]. These models are pre-trained on large corpuses of neural population data and then fine-tuned for downstream tasks such as behavior decoding and brain-computer interface (BCI) control [42], leading to improved performance and generalization to novel sessions and animals. While these models are a promising

step in the right direction, two crucial elements are currently missing. First, the pre-training is only performed on neural data from the sensorimotor network (M1, PMd, S1) which limits the applicability of these approaches to whole-brain analyses. Second, these models do not explicitly model the underlying brain regions, instead treating the population as a homogenous set of neurons. We argue that a foundation model for neural spiking data must be able to seamlessly translate across all spatial scales, including population-level, region-level, and single-neuron-level dynamics.

In this work, we build towards a first foundation model for neural spiking data which can solve a diverse set of predictive tasks across diverse brain areas. Similar to [41, 42], we utilize masked modeling where parts of the input are masked and then reconstructed using the unmasked inputs. To explicitly capture neural dynamics across all spatial scales, we introduce a multi-task-masking (MtM) approach where the model alternates between masking then reconstructing neural activity in masked *time steps*, *neurons*, and *brain regions*. We learn a "prompt" token which allows the model to seamlessly switch between different masking objectives during training [35]. During evaluation, this prompt token can be utilized to perform "mode switching" where downstream tasks are associated with specific masking schemes.

We evaluate our approach using the International Brain Laboratory (IBL) repeated site dataset [18] which consists of multi-region Neuropixels recordings that target the same brain regions (secondary visual areas, hippocampus, and thalamus) across multiple animals. We design a number of unsupervised and supervised prediction tasks which include single-neuron and region-level activity prediction, forward prediction, and behavior decoding. We benchmark our MtM approach against the temporal masking scheme used by Neural Data Transformer (NDT) [41] and the random token masking scheme used by Neural Data Transformer 2 (NDT2) [42]. We show that even with the same architecture, our MtM approach significantly outperforms the temporal masking baselines and enables multi-task learning. To demonstrate that our MtM approach is a viable recipe for large-scale pre-training, we train across 34 animals and fine-tune on 5 held-out animals. The performance of our MtM approach continuously scales with more training sessions, indicating its potential as a "universal translator" of neural dynamics at single-cell, single-spike resolution.

The contributions of this work include:

- A novel multi-task-masking (MtM) approach which can be applied to multi-region datasets to successfully learn representations that lead to better downstream task performance.

- A prompt-based approach for test-time adaptation which improves performance on a variety of downstream tasks during inference.

- Scaling results that demonstrate that having data from more animals provides benefits on held-out animals and sessions as well as on unseen tasks.

- A new multi-task, multi-region benchmark for evaluating foundation models of neural population activity.

## 2 Related Work

### 2.1 Foundation models for neuroscience

The advent of large-scale, self-supervised foundation models has marked a paradigm shift across various domains of machine learning. These models, diverging from traditional annotation-reliant supervised models, exhibit an impressive ability to generalize across a spectrum of tasks. These models have transformed natural language processing [27, 28, 40], vision [29, 16], and robotics [30], and are beginning to reshape the landscape of life sciences [5, 1]. The application of foundation models to neuroscience is of significant interest. While there has been considerable progress in building large-scale models for EEG, [6], fMRI [36], and sEMG [19], which have ample data availability, no large-scale model exists for neural data at single-neuron, single-spike resolution. To address this gap, two new methods, a supervised method, POYO [3], and a self-supervised method, NDT2 [42], were trained on a large corpus of spiking data from ∼12 monkeys. While promising, the datasets used for training are from just a few animals and brain regions, and therefore lack scale and diversity, limiting their applicability to other brain areas and behavioral contexts.

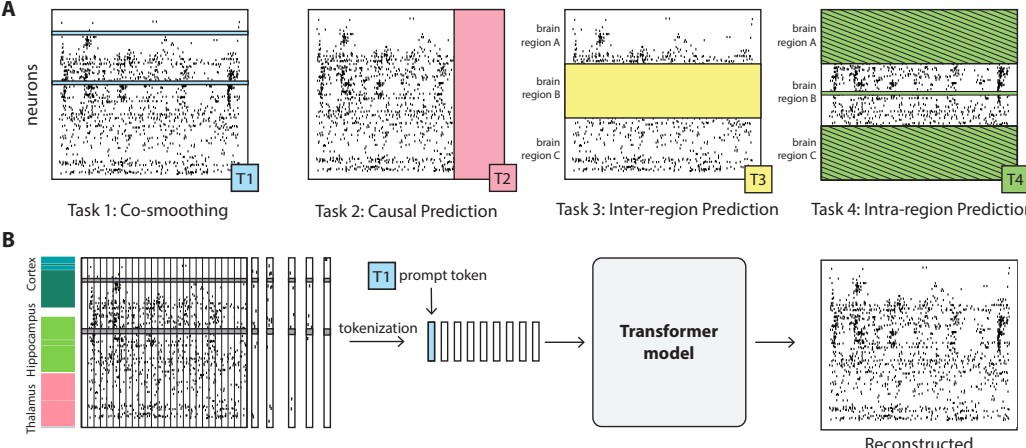

Figure 1: *Schematic illustration of our Multi-task-Masking (MtM) approach*: (A) We introduce four metrics for evaluating foundation models of neural population activity: neuron co-smoothing, causal prediction, inter-region prediction, and intra-region prediction. For each masking scheme, the colored area indicates what is masked and then reconstructed for evaluation. For intra-region prediction, the colored areas with hatched lines indicate areas which are masked, but not reconstructed for evaluation. Each metric can be associated with a specific masking scheme during training (T1, T2, etc.). (B) We alternate between different masking schemes during training along with a learnable "prompt" token which provides context to the model about the associated task [35]. During evaluation, we provide the associated prompt token for the downstream task to perform test-time adaptation of the model. Our MtM approach is architecture-agnostic as masking is performed on the input data (not the tokens). For a full discussion of MtM, see Section 3.

## 2.2 Transformer architectures for neural population activity

Transformers [39] have recently been applied to neural population activity in both supervised [3, 20] and self-supervised [41, 42] settings. POYO [3] is a supervised multi-session model for predicting behavior from neural activity. The POYO architecture utilizes learnable unit embeddings for each neuron and a novel approach for tokenizing individual spike events with relative position encodings to incorporate precise spike timings [3]. For self-supervised learning, NDT1 [41] and NDT2 [3] are two existing transformer-based methods. Both utilize masked modeling and assume a Poisson emission model. For NDT1, each time bin is a token with dimensionality equal to the number of neurons. NDT1 uses a simple encoder-only transformer to reconstruct masked time bins during training. To incorporate information across many sessions spanning different sets of neurons and across individuals, NDT2 [42] was introduced. This model is ViT-style [7] time-series transformer architecture that uses spatiotemporal patches as tokens and a learned session-level context embedding. A token is defined as a single time step of neural activity for a subset of neurons and is masked and reconstructed during training. By performing patching of the neurons, NDT2 can easily be applied to multiple sessions.

## 2.3 Multi-region models

How information is encoded within and across different brain areas is an important question that underlies the study of brain organization [12], the evolution of brain development [2], and the diagnosis of different network-level brain diseases [4]. Recent advances in electrophysiological techniques now allow for recording neural activity across many interconnected regions simultaneously [13, 33, 43, 37]. This has inspired recent efforts to build more fine-grained estimates of multi-region communication [26, 32, 12] and to investigate neuron-level information processing across multiple brain areas [17]. To analyze these multi-region datasets, generative models have been developed that aim to identify low-dimensional latent variables representing shared activity among recorded areas [11, 31, 14, 8, 9]. Recently, dCSFA [11] and DLAG [9] were introduced to model temporal delays between two brain areas. mDLAG [10] further extends this approach to an arbitrary number of brain areas. While these approaches are interpretable solutions to understanding intra- and inter-region

neural dynamics, they also make limiting assumptions on the structure of the communication signals and cannot be easily scaled to many brain regions and sessions.

# 3 Methods

A foundation model for neural activity must be able to seamlessly "translate" across all spatial scales of the brain, including population-level, region-level, and single-neuron-level dynamics. To this end, we introduce a multi-task-masking (MtM) approach for self-supervised learning of neural activity. During training, we alternate between masking and then reconstructing neural activity across masked *time steps*, *neurons*, and *brain regions*. We utilize a learnable "prompt" token which provides the model with context about which masking scheme is being applied during training. This prompt token can be passed to the model at test time to adapt the model to the associated downstream task [35].

## 3.1 Masking schemes

Masked modeling for neural activity is typically performed by masking and then reconstructing activity in random time steps [41, 42]. While this masking scheme allows for learning temporal dynamics, it can ignore important spatiotemporal structure present in neural activity. To address this limitation, we propose four masking schemes designed to capture diverse patterns in neural data (see Figure 1 for a visualization of these masking schemes).

- **Causal masking**. Similar to a GPT-like model [28], we mask future time steps and then predict them using past time steps. While we learn next time step prediction, this masking scheme can be extended to multiple future time steps prediction as well.
- **Neuron masking**. We randomly mask neurons and reconstruct their activity using the unmasked neurons as context. This masking scheme allows the model to learn how individual neurons relate to the activity of the full population. This is conceptually similar to "coordinated dropout" introduced in [15].
- **Intra-region masking**. We randomly mask neurons in a randomly chosen brain region and then reconstruct them using only unmasked neurons from the **same** region as context. This masking scheme allows the model to learn intra-area dynamics.
- **Inter-region masking**. We randomly mask neurons in a randomly chosen brain region and then reconstruct them using unmasked neurons in **other** regions as context. This masking scheme allows the model to learn about cross-region interactions.

Each of these masking schemes teaches the model about different structure in neural populations. We hypothesize that a model trained with **all** these diverse masking schemes will be able to solve many different tasks at inference time.

## 3.2 Multi-task-masking (MtM)

To train using MtM, we randomly sample a masking scheme $M$ for each batch of neural data. We mask the input data according to the sampled masking scheme and then pass this to a tokenizer and transformer-based architecture. To preserve the temporal or spatial order of the data, we add positional embeddings $PE$ to the neural data tokens $Z$. To provide context to the model about the masking scheme that was sampled, we prepend a learnable "prompt" token $P$ to the neural data tokens. This prompt token is a D-dimensional learnable embedding which the model can use to adapt its behavior during training or evaluation. For a batch of neural data $X$, our training is as follows:

$$
\begin{aligned}
&M \sim \mathcal{U}(\text{causal, neuron, intra-region, inter-region}) \\
&Z = \text{Tokenizer}(M \odot X) \\
&Z_{\text{pos}} = Z + PE \\
&Z_{\text{prompt}} = [P, Z_{\text{pos}}] \\
&r = \text{Transformer}(Z_{\text{prompt}}) \\
&\hat{X} \sim \text{Poisson}(X \mid r)
\end{aligned}
\tag{1}
$$

where $\hat{X}$ is the neural data reconstructed by the model, based on the time-varying rates inferred from the transformer's output, assuming a Poisson emission model. The MtM approach is agnostic to the

choice of tokenizer and transformer, allowing it to be utilized with any architecture. For this work, we utilize the same tokenization scheme and transformer-based architecture as NDT1 and NDT2 for all comparisons to these methods (see Section 2.2 for details).

### 3.3 Prompt-based test-time adaptation

Depending on the downstream task, the information learned using each masking scheme might be more or less useful. We utilize a prompt-based "mode switching" [35] approach where the prompt token that is best associated with the downstream task is prepended to the neural data tokens during inference and fine-tuning.

## 4 Evaluation

### 4.1 Dataset

For training and evaluating our models, we use the International Brain Laboratory repeated sites dataset [18]. This dataset consists of Neuropixels recordings collected from nine labs which utilize a standardized experimental pipeline. The recordings target the same five brain regions across 48 adult mice performing a complex decision-making task. The probe was localized after the experiments using reconstructed histology and the brain regions were annotated. We utilize trial-aligned, spike-sorted data from 39/48 mice for our analyses. From these recordings, we have a total of 26,376 neurons for training and evaluation (~676 neurons per session on average). We bin the neural activity using 20ms windows and we fix the trial-length to 2 seconds (200 time bins). For behavior decoding, we exclude trials based on reaction time outliers as defined by the IBL brain-wide map [17].

### 4.2 Metrics

We utilize a number of unsupervised and supervised metrics to evaluate how well a neural population model generalizes to different downstream predictive tasks.

- **Co-smoothing**. Predicting the activity of a held-out neuron using all other neurons [25].
- **Forward prediction**. Predicting future activity from past activity. We predict the last 10% (200 ms) of the trial-aligned activity (2 seconds) for this metric.
- **Intra-region co-smoothing**. Predicting the activity of a held-out neuron using neurons in the **same** brain region.
- **Inter-region co-smoothing**. Predicting the activity of a held-out neuron using neurons in **other** brain regions. This is similar to the leave-one-out region validation from [10].
- **Choice decoding**. Predicting the choice the mouse makes using trial-aligned neural activity.
- **Motion energy decoding**. Predicting motion energy of the mouse's whiskers using trial-aligned neural activity. The motion energy is extracted from simultaneous video data.

We evaluate each unsupervised activity prediction metric for all neurons in a session. To evaluate activity prediction, we utilize the co-bps metric introduced in [25] which measures the performance of a model using bits per spike; see the Appendix for a detailed definition. For behavior prediction, we train a linear classifier on the output firing rates of each model. To predict choice, we use the spiking activity of all neurons across all timesteps. To predict motion energy at each timestep, we again use the neural activity across all timesteps. We use accuracy as the metric for choice decoding and the R-squared metric to quantify the proportion of variance explained for decoding motion energy.

### 4.3 Metrics and masking

By design, there is a correspondence between the metrics introduced in 4.2 and the novel masking schemes introduced in 3.1. The correspondences are as follows: neuron masking and co-smoothing, causal masking and forward prediction, intra-region masking and intra region co-smoothing, and inter-region masking and inter-region co-smoothing. For each evaluation metric, we propose a masking scheme that should lead to good downstream performance. By alternating between these masking schemes during training and utilizing a learnable prompt token at inference time, MtM should be able to generalize well to these different evaluation tasks.

For choice decoding and motion energy decoding, the correct prompt token at inference time is unknown. For our experiments, we utilize the neuron masking prompt token for choice decoding and causal masking for motion energy prediction. We found, however, that the choice of prompt token for behavior decoding is not that important for good downstream performance (see Appendix C).

### 4.4 Test-time neural activity masking

Activity prediction benchmarking is traditionally performed using a pre-fixed set of held-out neurons [25]. Models learn to predict the same held-out neurons using the held-in neurons during training. While this evaluation scheme works for a fixed held-out dataset, evaluating activity prediction on all neurons and on all brain regions would require training hundreds to thousands of individual models. Since the goal of a foundation model is to have a single model perform well across all metrics, we argue this evaluation method must be changed.

Therefore, we introduce a novel test-time evaluation scheme for benchmarking foundation models of neural spiking activity. During training, all models are trained on the full neural population activity with model-specific learning schemes. During evaluation, we zero mask different parts of the input data to construct held-out subsets of data to evaluate the model. We utilize this test-time masking scheme to compute all the activity prediction metrics introduced in Section 4.2. Due to the masking of sub-token channels during co-smoothing, inter-region and intra-region tasks, we attempted to concatenate a binary mask to the model input to indicate which channels were masked. However, this approach did not yield any advantages for NDT1 compared to zero masking. We provide further details on this in the Appendix.

## 5 Experiments

### 5.1 Architectures

For all of our experiments, we implemented two existing transformer architectures designed for neural population activity. For single-session evaluation, we re-implemented NDT1 [41] and NDT2 [42]. For multi-session evaluation, we re-implemented NDT1-stitch and NDT2 (described in Section 2.2). NDT1-stitch learns, for each session, a linear projection layer for embedding the neural activity vector. We also include a session-level context embedding for NDT1-stitch as we found it improved multi-session training. NDT2 utilizes a ViT-style [7] transformer architecture and a learned session-level context embedding. We set the patch size of NDT2 to be 128 neurons as, with ~676 neurons on average per session, smaller patch sizes were prohibitively slow to train. For more details about the architectures and re-implementation details, see Appendix D.

### 5.2 Single-session

**Masking scheme ablation.** To understand the importance of each masking scheme, we first train and evaluate all architectures on a single session using each individual masking scheme introduced in Section 3.1. We also train and evaluate a MtM model with and without prompting. As discussed in Section 4.3, each masking scheme should perform best on the associated metric and MtM should perform well across all metrics.

**MtM vs. temporal masking.** For a comprehensive evaluation of the MtM approach in comparison to temporal masking, we then train the single-session architectures with temporal mask-

Table 1: *The performance of single-session NDT1 trained with various masking schemes on neural activity reconstruction tasks.* The metrics are in units of bits per spike (bps), averaged across all neurons in one session. A higher bps value indicates better performance.

| Masking | Activity Reconstruction | | | |
| --- | --- | --- | --- | --- |
| | Co-Smooth | Forward Prediction | Intra-Region | Inter-Region |
| Temporal (Baseline) | 0.84 | 0.42 | -0.20 | 0.57 |
| Neuron | **1.04** | -0.21 | -0.22 | 0.78 |
| Causal | 0.44 | 0.48 | -0.36 | 0.23 |
| Intra-Region | -9.86 | -2.97 | 0.32 | -9.06 |
| Inter-Region | 0.92 | 0.01 | -0.58 | **0.90** |
| MtM (Not Prompted) | 0.99 | 0.54 | 0.42 | 0.83 |
| MtM (Prompted) | 0.98 | **0.57** | **0.43** | 0.84 |

ing and MtM across all 39 sessions (animals). Each model is trained for 1000 epochs, with the best checkpoint selected for evaluation based on the highest average single-neuron reconstruction $R^2$ across the 50 most active neurons in a session. We fix the learning rate and model architectures for all experiments. We evaluate the trained models on all metrics introduced in Section 4.2. For additional details about the hyperparameters and baselines, see Appendix D.

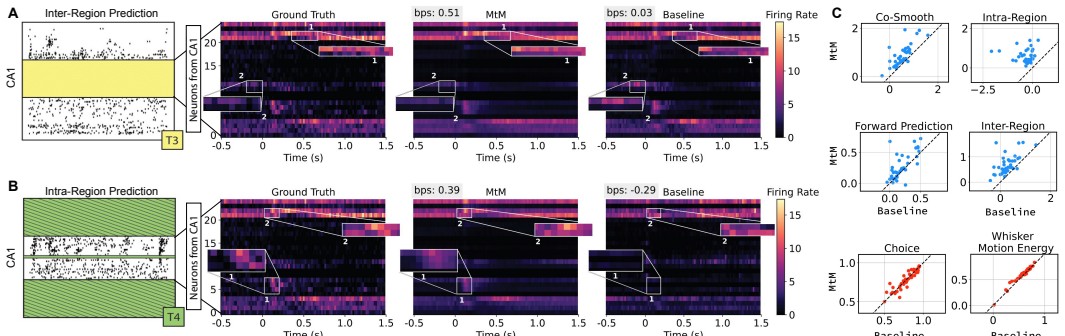

Figure 2: *Comparison of the temporal masking baseline and our proposed MtM method on single-session data.* (A) and (B) show trial-averaged raster maps of CA1 for ground-truth data, MtM, and the temporal baseline. (A) The predictions from MtM and the temporal baseline are after inter-region masking where neurons in CA1 are predicted from all **other** brain regions. We highlight two areas (1 and 2) where MtM shows qualitatively better predictions of activity. (B) The predictions from MtM and the temporal baseline are after intra-region masking where all neurons in CA1 are predicted from other neurons in the **same** brain region. We again highlight two areas (1 and 2) where MtM shows qualitatively better predictions of activity. (C) Activity reconstruction and behavior decoding across 39 sessions for MtM and temporal masking. Each point represents one session. For activity reconstruction, we report the average bps. For choice and whisker motion energy decoding, we report the average accuracy and $R^2$, respectively, across all test trials. We use the NDT1 architecture for all comparisons.

## 5.3 Multi-session

**MtM vs. temporal masking pre-training.** We are also interested in evaluating the performance of MtM, in comparison to temporal masking, for multi-session pretraining. Although the identity of the neurons is changing across sessions, we hypothesize that by training across multiple IBL repeated site datasets, which share anatomical structure, our MtM-based approach should generalize better to unseen IBL repeated site sessions. To this end, we pretrain all multi-session architectures using MtM and temporal masking on 10 and 34 sessions of data. We then evaluate these multi-session models on 5 held-out sessions by fine-tuning their self-supervised loss (MtM or temporal) on each held-out session. This allows the models to learn session-specific information such as the session embeddings. We report all metrics across these 5 held-out sessions which include 397-579 trials in the training split, 57-83 trials in the validation split, and 114-166 trials in the test split.

**Behavior decoding from individual brain regions.** To evaluate how well our fine-tuned, 34-session pre-trained MtM can generalize to unseen tasks, we also perform behavior decoding using single brain regions from the 5 held-out sessions. To perform behavior decoding using a single brain region, we mask out all other regions and predict the rates of the neurons in the specified region. Then, we train a linear model on these output rates to predict choice and whisker motion energy. For MtM, we prepend the intra-region prompt token as it is associated with this downstream task. We compare behavior decoding results for our 34-session pre-trained MtM to the 34-session pre-trained temporal masking model across all brain regions.

## 6 Results

For all results in the main text, we utilize NDT1 and NDT1-stitch architectures. We found that the NDT1 architecture outperformed the NDT2 architecture across all metrics and sessions for both temporal masking and MtM. We hypothesize that the patching scheme of NDT2 does not generalize well to spike-sorted, multi-region Neuropixels recordings which have up to ∼676 neurons on average per session. All NDT2 results are reported in Appendix B and C. These results have similar trends (with overall lower metric scores) as the NDT1 results.

### 6.1 Single-session

**Masking scheme ablation.** The results for the masking scheme ablation on a single session are shown in Table 1. We report the neuron-averaged activity metrics for all masking schemes, including the co-smoothing, forward prediction, intra-region, and inter-region activity prediction. As shown in the table, each masking scheme leads to an improvement on its associated metric (see Section 4.3)

over the temporal masking baseline. This is a promising result as it illustrates how each masking scheme can teach the model about a different aspect of the neural population. Our MtM method also shows strong improvements in activity prediction across all 4 metrics in comparison to the temporal baseline. Although the activity prediction results of the single masking schemes can sometimes outperform MtM on the associated metric, the overall performance of MtM across all metrics is high. This demonstrates how training with diverse masking schemes can lead to a more structured understanding of neural activity. Finally, we show that MtM with prompting is a modest (in 3/4 metrics) improvement over MtM without prompting. Overall, this masking scheme ablation demonstrates the strength of our MtM approach for structured learning of neural data.

**MtM vs. temporal masking.**    We show results for our comparison of MtM to temporal masking across all 39 sessions in Figure 2. As seen in Figure 2, our MtM training approach leads to significant improvements across all 4 unsupervised activity prediction tasks. The largest improvements of MtM over temporal masking are for intra and inter-region activity prediction, as temporal masking is unable to learn this structure. For behavior decoding, we find that MtM and temporal masking have comparable results for choice decoding and MtM slightly outperforms temporal masking on whisker motion energy prediction. As we are using the full population of neurons for behavior decoding in these analyses, the similarity in results between MtM and temporal masking is unsurprising given that the temporal masking is performed on the full population for each time step. We hypothesize that when decoding behavior from individual brain regions, the MtM approach should outperform temporal masking as it learns brain region-specific structure (see Figure 5). These single session results demonstrate that MtM is a promising method for learning population-level, region-level, and single-neuron-level dynamics from neural population data.

## 6.2    Multi-session

**MtM vs. temporal masking pre-training.** We report results for multi-session pretraining using MtM and temporal masking on 34 sessions of data in Table 2 and Figure 3. In Table 2, we show session-averaged results on the 5 held-out sessions for both the single-session and 34-session pre-trained MtM and temporal masking. For both single-session and multi-session, MtM outperforms temporal masking across all metrics except choice decoding (where the results are quite similar). Both methods benefit from multi-session pre-training as all unsupervised and supervised metrics improve for the 34-session pretrained models. Similar to the single-session results, the biggest improvements for MtM are for the unsupervised activity metrics especially inter- and intra-region prediction.

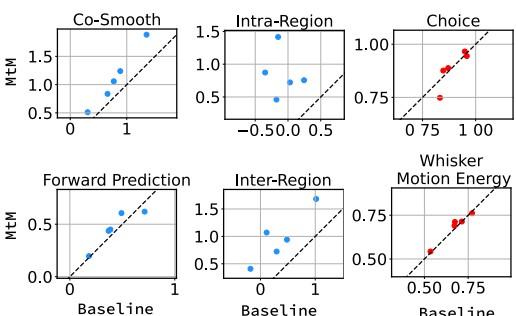

Figure 3: *Fine-tuning performance comparison of NDT1-stitch pretrained with MtM vs. temporal masking for activity reconstruction and behavior decoding across 5 held-out sessions.* For activity reconstruction, each point shows the average bps across all neurons in a held-out session. For behavior decoding, each point shows the trial-averaged accuracy (choice) and $R^2$ (WME).

In Figure 3, we show a scatter plot of all metrics for the 5 held-out datasets for MtM and the temporal baseline. MtM shows improvement over the temporal baseline for all activity metrics.

Table 2: *Fine-tuning performance of NDT1-stitch pretrained with MtM vs. temporal masking on activity reconstruction and behavior decoding.* Activity reconstruction performance is reported in neuron-averaged bps. For behavior decoding, trial-averaged accuracy and $R^2$ are shown for choice and whisker motion energy, respectively. All metrics are averaged across 5 held-out sessions, and a higher value indicates better performance.

| Training | Masking | Activity Reconstruction | | | | Behavior Decoding | |
|---|---|---|---|---|---|---|---|
| | | Co-Smooth | Forward Pred | Intra-Region | Inter-Region | Choice | Whisker Motion Energy |
| Single-Session | Temporal (Baseline) | 0.55 | 0.17 | -0.49 | 0.18 | **0.87** | 0.65 |
| | MtM (Prompted) | **1.00** | **0.28** | **0.70** | **0.83** | 0.85 | **0.67** |
| Multi-Session | Temporal (Baseline) | 0.79 | 0.43 | -0.08 | 0.34 | **0.89** | 0.67 |
| | MtM (Prompted) | **1.11** | **0.46** | **0.85** | **0.96** | 0.88 | **0.68** |

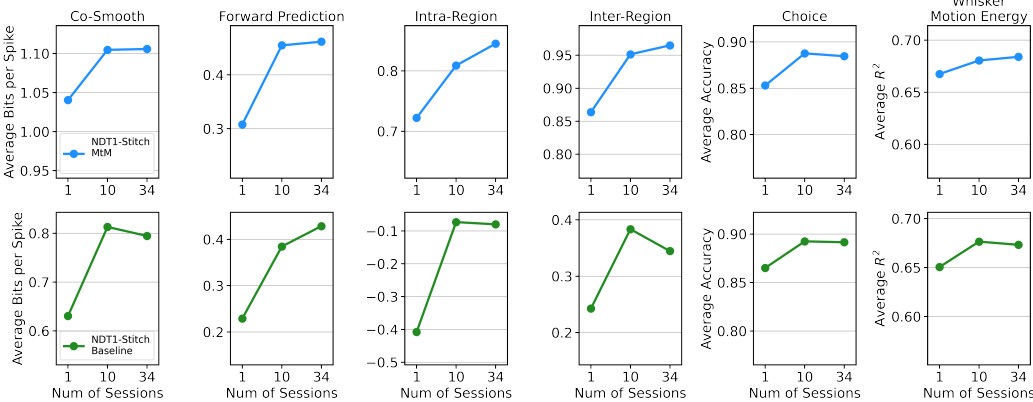

Figure 4: *Comparison of scaling curves between NDT1-stitch pretrained with the MtM method vs. the temporal masking baseline.* The reported metrics - neuron-averaged bits per spike (bps), choice decoding accuracy, and whisker motion energy decoding $R^2$ - are averaged over all 5 held-out sessions. We fine-tune each pretrained model with its self-supervised loss (MtM or temporal) on the 5-heldout sessions and then evaluate with all of our metrics. "Num of Sessions" denotes the number of sessions used for pretraining.

The results of our scaling analysis (1, 10, and 34 session training), can be seen in Figure 4. For MtM, scaling the number of pretraining sessions leads to improved performance on all downstream metrics except for choice decoding which saturates at 10 sessions. For temporal masking, the performance of co-smoothing, intra-region, inter-region, choice decoding, and motion energy prediction saturates at 10 session pretraining. These results demonstrate that the MtM is a more promising approach than temporal masking for scaling neural population models to multi-animal, multi-regional datasets.

**Behavior decoding from individual brain regions.** The results for behavior decoding using single brain regions on the 5 held-out sessions are shown in Figure 5. We provide the chance level for each session in the Appendix. From these results, one can see that MtM provides a significant improvement on brain region behavior decoding in comparison to temporal masking across both choice decoding and whisker motion energy prediction. This is especially apparent in hippocampus brain areas (orange) where for session 5dee0eb, the temporal masking baseline is below chance-level on choice decoding for many sub-areas, but MtM is above chance level on all sub-areas. These results illustrate the ability of MtM to extract region-specific information from multi-region neural populations.

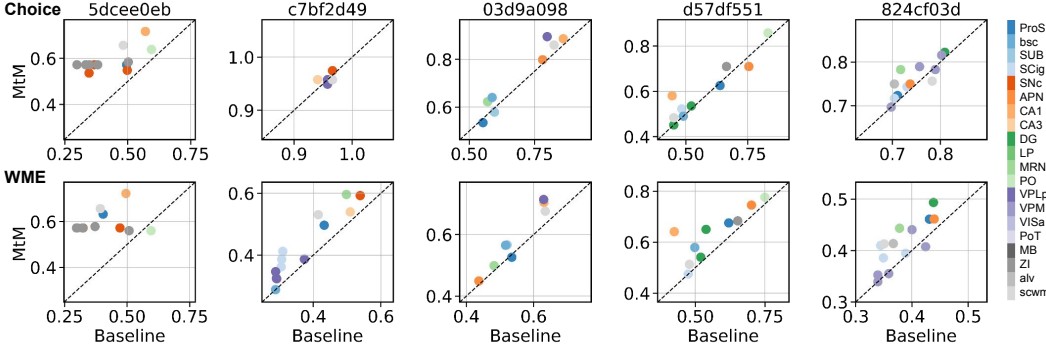

Figure 5: *Comparison of NDT1-stitch pretrained with the MtM method vs. the baseline temporal masking on behavior decoding from individual brain regions.* The rows display choice decoding accuracy and whisker motion energy decoding $R^2$. Columns represent individual held-out sessions. Each point shows the behavior decoding performance when using neural activity from a specific brain region, with colors denoting different brain regions.

# 7 Discussion

In this work, we take an important step toward a foundation model of the brain at single-cell, single-spike resolution. We introduce a novel approach to self-supervised learning, multi-task-masking (MtM), which is able to learn population-level, region-level, and single-neuron level structure from neural population activity. We validate this approach on neural recordings taken from a large, diverse dataset (39 mice across multiple brain regions). We find that MtM significantly outperforms current state-of-the-art masking modeling schemes for neural data prediction, and enables multi-task generalization. We also provide scaling results demonstrating that MtM is a promising approach for large-scale pre-training on neural population data across animals and sessions.

A number of limitations remain. First, the data diversity, while higher than other neural pre-training datasets [3, 42], is still significantly lower than a full brain-wide map of neural population activity [17]. Training MtM on the IBL brain-wide map dataset, which contains 300 brain regions and hundreds of animals, is an exciting future direction. Second, our current architectures, NDT1 and NDT2, are simple time-series transformers that utilize basic tokenization schemes, i.e. temporal tokens. Recent work has demonstrated that learning global dependencies across temporal tokens leads to poor forecasting results on multivariate time-series datasets [21] and that more sophisticated architectures can outperform the NDT architectures on behavior decoding from neural data [3]. Therefore, improving the underlying architecture of MtM is another direction that should be explored.

## Acknowledgments and Disclosure of Funding

This project was supported by the Wellcome Trust (209558 and 216324), National Institutes of Health (1U19NS123716), the Simons Foundation, the National Science Foundation (NSF award CIF:RI:2212182, NSF CAREER awards IIS-2146072), NSERC (Discovery Grant: RGPIN-2020-05105; Discovery Accelerator Supplement: RGPAS-2020-00031; Arthur B. McDonald Fellowship: 566355-2022) and CIFAR (Canada AI Chair; Learning in Machine and Brains Fellowship), and by DoD OUSD (R&E) under Cooperative Agreement PHY-2229929 (The NSF AI Institute for Artificial and Natural Intelligence), as well as generous gifts from the Alfred Sloan Foundation, the McKnight Foundation, and the CIFAR Azrieli Global Scholars Program.

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

# Appendix

## A    Additional results

**Architecture agnosticity.**    We have added an experiment where we trained the LFADS (a sequential VAE) architecture [24] on 5 test sessions using the temporal masking baseline and our MtM learning objective. The results of the LFADS analysis are shown in Table 3. We found that the LFADS architecture trained with MtM outperforms temporal masking on all activity prediction tasks and is competitive with NDT1 trained with MtM (although slightly worse). Behavior prediction is comparable between MtM and temporal masking. This experiment suggests that MtM is an architecture-agnostic learning objective that can be used with both transformers and RNNs.

Table 3: *The performance of single-session NDT1 vs. LFADS trained with various masking schemes on activity reconstruction and behavior decoding tasks.* The metric for activity reconstruction is in units of bits per spike (bps), averaged across all neurons in one session. Behavior decoding is assessed using accuracy for choice and $R^2$ for whisker motion energy. For all metrics, a higher value indicates better performance.

|  | Activity Reconstruction | | | | Behavior Decoding | |
| --- | --- | --- | --- | --- | --- | --- |
|  | Co-Smooth | Forward Pred | Intra-Region | Inter-Region | Choice | Whisker Motion Energy |
| NDT1 Baseline | 0.55 | 0.17 | -0.49 | 0.18 | 0.87 | 0.65 |
| NDT1 MtM | 1.00 | 0.28 | 0.70 | 0.83 | 0.85 | 0.67 |
| LFADS Baseline | 0.86 | 0.09 | -0.34 | 0.50 | 0.86 | 0.67 |
| LFADS MtM | 0.87 | 0.26 | 0.65 | 0.76 | 0.82 | 0.67 |

**Estimation of "functional connectivity".**    We utilized our pretrained MtM model to ask how well individual brain areas predict each other. We fine-tuned our pretrained MtM model on each test session with a new inter-region task where we randomly dropout some of the regions when predicting one region. Then, we used MtM to predict region A from region B for all regions in the session. The prediction matrix is shown in Figure 6. Even without explicit training to predict one region from another region, there is interesting structure in the matrix as some regions predict each other well, for example in visual areas. We also fine-tuned our pretrained MtM model on each test session to predict region A from region B for 5 selected regions. We compared its performance to NDT1 regression models trained from scratch for the same region-to-region predictions. In Figure 7, the prediction matrices show that the fine-tuned MtM outperforms the NDT1 regression models in region-to-region prediction. Taken together, these results suggest that MtM can be used to explore the "functional connectivity" of different areas.

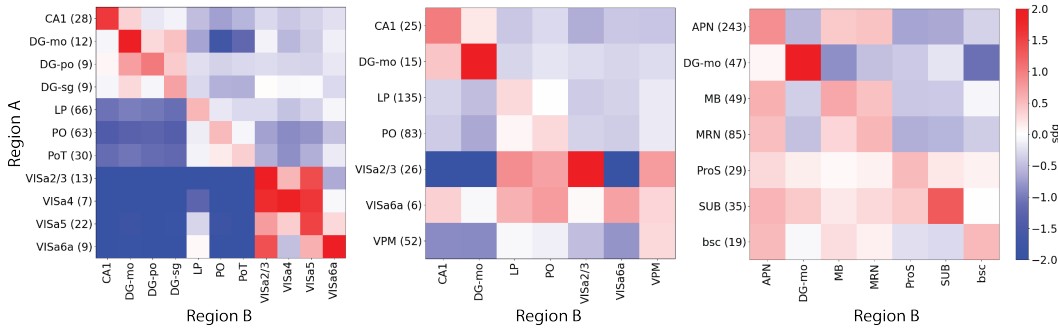

Figure 6: *Predicting region A from region B matrix using MtM.* Each matrix is a test session and each entry is the co-bps when predicting region A from region B using MtM. We fine-tune MtM on each session using an inter-region masking scheme where we randomly dropout regions to encourage the model to utilize every region. We filter out all regions with less than 5 neurons.

**Cross-species transfer.**    While the IBL repeated site dataset used for training MtM has high diversity of brain regions, it lacks task (species) diversity. To demonstrate that MtM can be a useful pretraining strategy for new tasks and species, we fine-tuned our 34-session pretrained NDT1-MtM model on the Neural Latents Benchmark (NLB) [25] MC_RTT dataset. This dataset consists of spiking activity

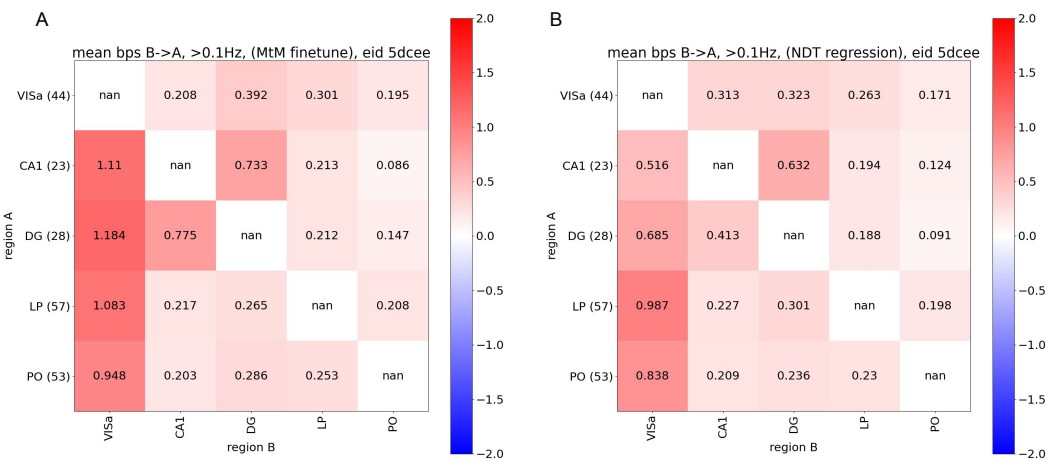

Figure 7: *A comparison of the pretrained MtM fine-tuned for region-to-region prediction vs. the NDT1 regression model trained from scratch for the same task.* The matrix in panel A displays region-to-region predictions from the fine-tuned MtM, while panel B presents the predictions from the NDT1 regression model. Both matrices are from the same session, with each entry representing the co-bps when predicting region A from region B.

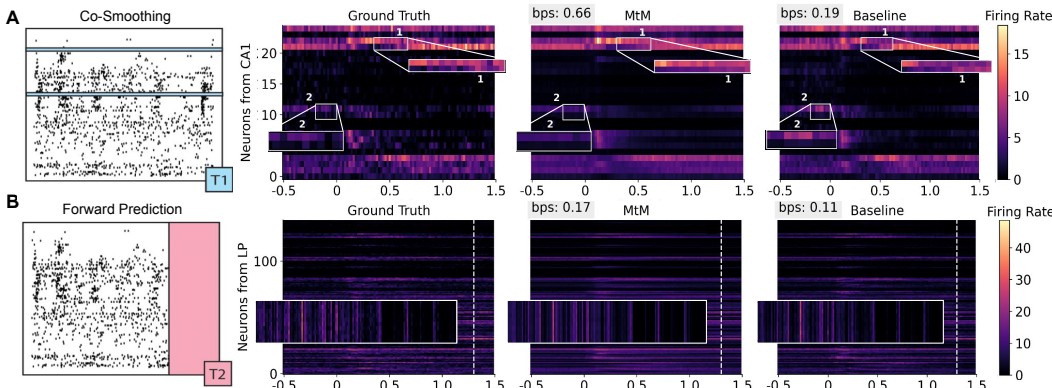

Figure 8: *Rastermap visualizations of MtM and temporal masking for a single session on co-smoothing and forward prediction tasks.* Each row corresponds to a different predictive task. The far left column is schematic of the predictive task, the second column is the ground-truth spiking activity in a region, and the third and fourth columns are predicted rates from MtM and temporal masking (baseline), respectively.

recorded from the primary motor cortex of a monkey performing a self-paced reaching task. We divided this dataset into two "brain regions" based on the NLB held-in and held-out neuron split. We then fine-tuned our model using the MtM learning objective. The MtM performance (0.19 bps) significantly improved upon NDT1's previous best performance (0.16 bps).

**Additional rastermap visualizations.** We have added visualizations of the rastermaps for NDT1 trained with MtM and temporal masking for each predictive task: neuron, causal, inter-region, and intra-region prediction. As can be seen in both Figure 2 and Figure 8, MtM leads to better predicted rastermaps for inter-, intra-, and neuron masking. For forward prediction, the improvement is more subtle, largely because temporal masking performs well at temporal predictive tasks. Overall, these visualizations give us confidence that MtM is a significant improvement over the baseline.

# B    Single-session details

**Masking scheme ablation.** To understand the importance of each masking scheme, we evaluate NDT1 and NDT2 trained with various masking schemes, including temporal, neuron, causal, intra-

Table 4: *The performance of single-session NDT1 trained with various masking schemes on activity reconstruction and behavior decoding tasks.* The metric for activity reconstruction is in units of bits per spike (bps), averaged across all neurons in one session. Behavior decoding is assessed using accuracy for choice and $R^2$ for whisker motion energy. For all metrics, a higher value indicates better performance.

| Masking | Activity Reconstruction | | | | Behavior Decoding | |
| --- | --- | --- | --- | --- | --- | --- |
| | Co-Smooth | Forward Pred | Intra-Region | Inter-Region | Choice | Whisker Motion Energy |
| Temporal (Baseline) | 0.84 | 0.42 | -0.20 | 0.57 | 0.66 | 0.56 |
| Neuron | **1.04** | -0.21 | -0.22 | 0.78 | 0.68 | 0.60 |
| Causal | 0.44 | 0.48 | -0.36 | 0.23 | **0.75** | 0.59 |
| Intra-Region | -9.86 | -2.97 | 0.32 | -9.06 | 0.55 | 0.38 |
| Inter-Region | 0.92 | 0.01 | -0.58 | **0.90** | 0.52 | 0.59 |
| MtM (Not Prompted) | 0.99 | 0.54 | 0.42 | 0.83 | 0.69 | 0.61 |
| MtM (Prompted) | 0.98 | **0.57** | **0.43** | 0.84 | 0.63 | **0.61** |

Table 5: *The performance of single-session NDT2 trained with various masking schemes on activity reconstruction and behavior decoding tasks.* The metric for activity reconstruction is in units of bits per spike (bps), averaged across all neurons in one session. Behavior decoding is assessed using accuracy for choice and $R^2$ for whisker motion energy. For all metrics, a higher value indicates better performance.

| Masking | Activity Reconstruction | | | | Behavior Decoding | |
| --- | --- | --- | --- | --- | --- | --- |
| | Co-Smooth | Forward Pred | Intra-Region | Inter-Region | Choice | Whisker Motion Energy |
| Random Token (Baseline) | -6.94 | 0.50 | -0.43 | -6.95 | 0.74 | 0.58 |
| Neuron | 0.91 | 0.18 | -0.26 | 0.62 | 0.65 | 0.62 |
| Causal | 0.02 | 0.52 | -0.42 | -0.20 | 0.69 | 0.59 |
| Intra-Region | -10.10 | -1.30 | 0.21 | -8.17 | 0.65 | 0.43 |
| Inter-Region | 0.63 | 0.18 | -0.63 | 0.66 | **0.75** | 0.39 |
| MtM (Not Prompted) | 0.90 | **0.56** | **0.47** | 0.80 | 0.68 | **0.62** |
| MtM (Prompted) | **0.92** | 0.54 | 0.46 | **0.81** | 0.69 | 0.62 |

region, inter-region, as well as the proposed MtM method with and without the prompt token on a selected single session. For both NDT1 and NDT2, Tables 4 and 5 show that the prompted MtM model outperformed the baseline temporal (random token) masking model on most evaluation tasks, except for choice decoding. In addition, each masking scheme performed well on its corresponding task. Figure 9 compares the baseline random token masking NDT2 to the prompted MtM NDT2 across 39 single sessions. The prompted MtM NDT2 consistently outperformed the baseline random token masking NDT2 on all activity reconstruction tasks, while performing similarly on behavior decoding tasks.

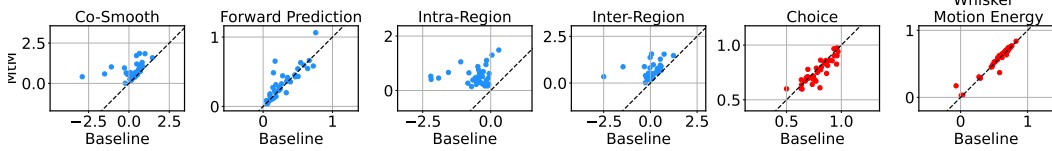

Figure 9: *Comparison of the random token masking baseline and the proposed MtM method for NDT2 on activity reconstruction and behavior decoding across 39 sessions.* For activity reconstruction, each point shows the average bps across all neurons in one session. For choice (whisker motion energy) decoding, each point represents the average accuracy ($R^2$) across all test trials in one session.

**Binary mask token ablation for NDT2.** To optimize NDT1 and NDT2 for predicting held-out neurons, sub-token channels are masked during co-smoothing, inter-region and intra-region tasks, except for forward prediction. Initially, a binary mask was concatenated to the input of NDT1 to indicate which channels were masked, but this approach did not show benefits, leading to the use of zero masking instead. We also tested concatenating a binary mask with NDT2, which showed a slight improvement in Table 6, though overall performance metrics for random token masking remained negative.

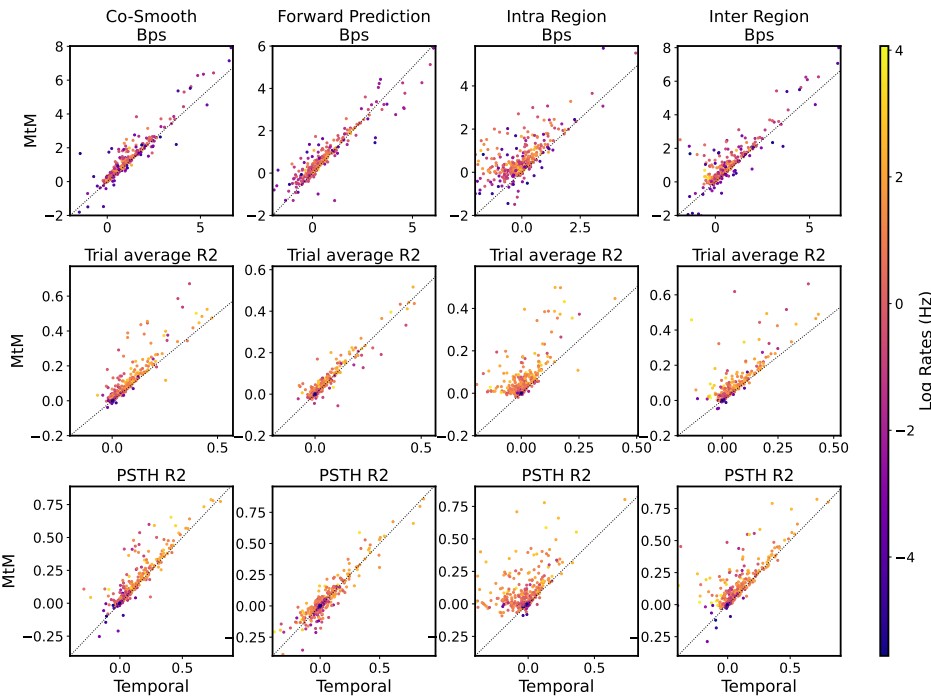

Figure 10: *Single neuron activity reconstruction analysis for NDT1 in one session.* To evaluate the reconstruction quality for each neuron, multiple metrics are computed: Bits per spike (Bps), $R^2$ between the ground truth and predicted peristimulus time histogram (PSTH $R^2$), and the single-trial $R^2$ averaged across all trials (Trial average $R^2$). Each point represents one neuron, with the color indicating the neuron's log firing rates in Hertz (Hz).

Table 6: *Ablation study on the impact of using a binary mask token with NDT2 for downstream activity reconstruction tasks.* Results are based on a single selected test session.

|  | Co-Smooth | Forward Pred | Intra-Region | Inter-Region |
|---|---|---|---|---|
| NDT2 Random Token | -6.84 | 0.40 | -0.84 | -6.39 |
| NDT2 Random Token + Binary Mask Token | -2.27 | 0.41 | -0.55 | -2.10 |
| NDT2 MtM | 0.85 | 0.51 | 0.32 | 0.73 |

**Single neuron evaluation.** To better understand neural activity prediction performance at a single-neuron level, we conducted an evaluation of single-session NDT1 on each neuron using bits per spike (bps), $R^2$ between the ground truth and predicted peristimulus time histogram (PSTH $R^2$), and the single-trial $R^2$ averaged across all trials (trial average $R^2$), on one session, in Figure 10. We find that MtM outperforms the temporal baseline across most neurons regardless of firing rate. We did find a strong correlation between the performance evaluated on each metric and the mean firing rates of each single neuron. For the bps (bits per spike) metric, scores for active neurons tend to be more concentrated, while scores for inactive neurons are relatively dispersed, exhibiting both extremely low and high values. For both $R^2$ metrics, the performance shows a positive correlation with the mean firing rates. In particular, those neurons with extremely low mean firing rates typically exhibited $R^2$ scores that were extremely low (approaching zero). This observation might be related to the inherent difficulty in predicting the behavior of neurons with low mean firing rates, or the property of metrics themselves. For instance, when applied to neurons with low mean firing rates, the $R^2$ metric might tend to yield values closer to zero. Across all three metrics (Bps/Trial average $R^2$/PSTH $R^2$), our model demonstrated substantial improvements for neurons with relatively higher mean firing rates. However, for neurons with lower mean firing rates, notable improvements were only observed in the bps metric.

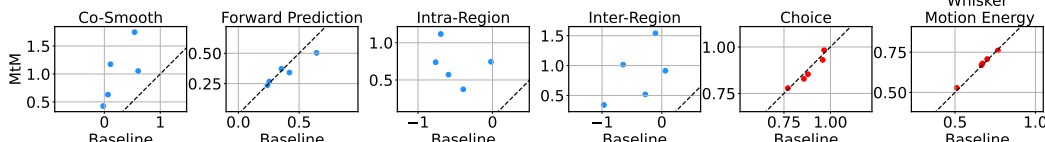

Figure 11: *Fine-tuning performance comparison of NDT2 pretrained with MtM vs. random token masking for activity reconstruction and behavior decoding across 5 held-out sessions.* For activity reconstruction, each point shows the average bps across all neurons in a held-out session. For choice (whisker motion energy) decoding, each point represents the average accuracy ($R^2$) across all test trials in one session.

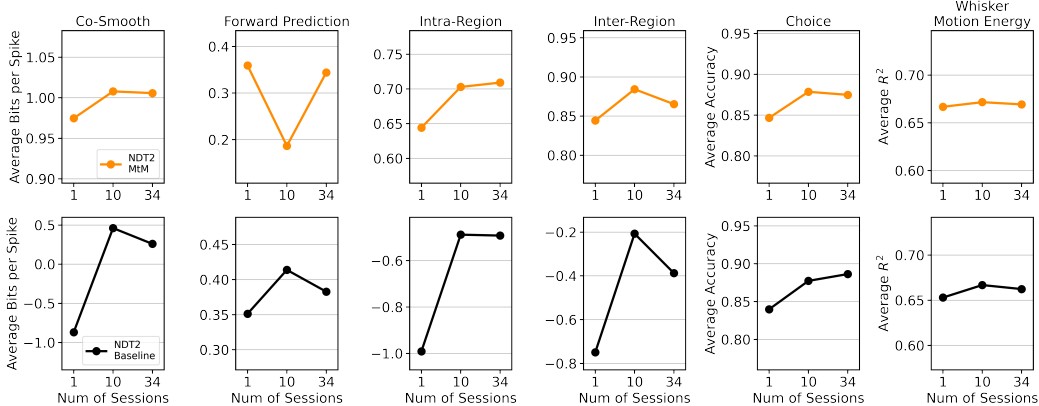

Figure 12: *Comparison of scaling curves between NDT2 pretrained with the MtM method vs. the random token masking baseline.* The reported metrics - neuron-averaged bits per spike, choice decoding accuracy, and whisker motion energy decoding $R^2$ - are averaged over all 5 held-out sessions used for fine-tuning on both activity reconstruction and behavior decoding tasks. "Num of Sessions" denotes the number of sessions used for pretraining.

## C   Multi-session details

For NDT2, we report results for pretraining using MtM and the baseline random token masking on 34 sessions of data in Table 7 and Figure 11. In Table 7, we show session-averaged results on the 5 held-out sessions for both the single-session and 34-session pre-trained MtM and random token masking. For both single-session and multi-session, MtM outperforms random token masking across all metrics except choice decoding (where the results are quite similar). Both masking schemes benefit from multi-session pre-training as all unsupervised and supervised metrics improve for the 34-session pretrained models. Similar to the single-session results, the biggest improvements for MtM are for the unsupervised activity metrics, especially inter- and intra-region prediction. In Figure 11, we show a scatter plot of all metrics for the 5 held-out datasets for MtM and the random token baseline.

Table 7: *Fine-tuning performance of NDT2 pretrained with MtM vs. random token masking on activity reconstruction and behavior decoding.* Activity reconstruction performance is reported in neuron-averaged bps. For behavior decoding, trial-averaged accuracy ($R^2$) for choice (whisker motion energy) decoding is shown. All metrics are averaged across 5 held-out sessions, and a higher value indicates better performance.

| Training | Masking | Activity Reconstruction | | | | Behavior Decoding | |
|---|---|---|---|---|---|---|---|
| | | Co-Smooth | Forward Pred | Intra-Region | Inter-Region | Choice | Whisker Motion Energy |
| Single-Session | Random Token (Baseline) | -0.87 | 0.35 | -0.99 | -0.75 | 0.84 | 0.65 |
| | MtM (Prompted) | **0.97** | **0.36** | **0.64** | **0.84** | **0.85** | **0.67** |
| Multi-Session | Random Token (Baseline) | 0.26 | 0.38 | -0.49 | -0.39 | **0.89** | 0.66 |
| | MtM (Prompted) | **1.01** | **0.34** | **0.71** | **0.87** | 0.87 | **0.67** |

**Scaling analysis.** To examine NDT2's ability of scaling data, Figure 12 shows that NDT2 multi-session pre-training also benefits from scaling from 1 to 10 sessions, but we only observe marginal gains or no improvement going from 10 to 34 sessions. NDT2 (Figure 12) benefits less from multi-session IBL pre-training compared to NDT1-stitch (Figure 4), likely due to the inability of NDT2 with a fixed patch size to handle the large neuron count variations (200 to 1000 neurons) across IBL sessions. Another reason is the NDT2 takes too many neuron patches at the same time, and it's very challenging to deal with this long sequences when data is scaling.

**Prompting mode ablation.** We also conducted ablation studies on NDT1-stitch of different prompt mode during behavior decoding. We only apply prompt during the model inference, and observe the different prompt effects to our behavior results. As shown in the Table 8, we ablate four prompt modes.

Table 8: *Evaluation of NDT1's behavior decoding performance when pretrained and fine-tuned using the MtM approach, tested with different prompt tokens at inference time.* Behavior decoding is assessed using trial-averaged accuracy for choice and trial-averaged $R^2$ for whisker motion energy, with the reported metrics averaged over 5 held-out sessions. For both metrics, a higher value indicates better performance.

| Prompting | Behavior Decoding | |
|---|---|---|
| | Choice | Whiker Motion Energy |
| Neuron | 0.88 | 0.68 |
| Causal | **0.89** | 0.68 |
| Intra-Region | 0.88 | 0.68 |
| Inter-Region | 0.88 | 0.68 |

# D   Experiment details

**Architecture details.** We created a schematic for the NDT1 and NDT2 architectures in Figure 13 to clarify how these architectures tokenize and reconstruct neural data. NDT1 uses the timestep for the positional encoding. NDT2 uses 2 types of position encoding: (1) timesteps and (2) neuron group identity.

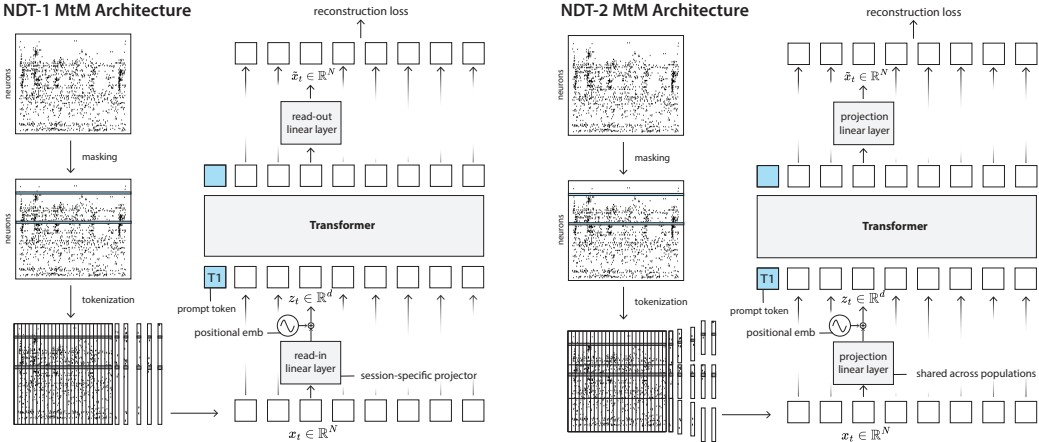

Figure 13: *Architectures for NDT1 and NDT2.* Each architecture tokenizes using temporal tokens although NDT1 uses the full population and NDT2 uses a patch of neurons. Following a linear transformation of the token, a position embedding is used before passing into the transformer. The output of the transformer is transformed back into neural activity using linear readout weights.

**Training details.** The 34-session NDT1 model has 25.55 million parameters, many of which come from session-specific stitchers. However, these stitchers use only one linear layer per session, making the model capacity equivalent to that of a single-session NDT1 model. We do not provide model

sizes for the single-session NDT1, as their parameter counts depend on the number of neurons in each session. Both the single-session and 34-session NDT2 models do not require session-specific stitchers and consist of 1.09 million parameters. We trained our model using AdamW optimizer [22] for 1000 epochs with a learning rate of $1e^{-4}$ using a cosine scheduler. We put a weight decay $0.01$ to avoid overfitting. We utilized a relatively small batch size of 16 during the training. We split our dataset based on the session to training, validation, and test set with a proportion of 70%, 10%, and 20%. We saved the model checkpoint based on a trial-average $R^2$ of top-50 active neurons, which we selected top-50 active neurons and calculated each neuron's $R^2$ through averaging the trials.

**Compute.** NDT1-stitch was trained on a machine with a single RTX8000 GPU. NDT2 was trained using Tesla V100 GPUs with 32Gb memory. The 10-session and 34-session NDT1-stitch models were trained for 1 and 3 days, respectively, while the 10-session and 34-session NDT2 models took 2 and 5 days, respectively. Single-session NDT1 and NDT2 models, as well as finetuning, were trained on a single Tesla V100 GPU, requiring 3 to 6 hours. We make sure our experiments are reproducible by seeding.

**Hyperparameters details.** The masking ratio is an important model hyperparameter for NDT1 and NDT2. For neuron, intra-region, temporal, and our proposed MtM masking schemes, the masking ratio is fixed at 30%, favoring the performance of the baseline temporal masking method across the activity reconstruction tasks. The causal (next timestep prediction) and inter-region (mask whole region) schemes do not have this hyperparameter, making their performance invariant to the selected mask ratio.

For NDT2, the spatiotemporal patch size is another important hyperparameter. Due to computational constraints, we set it to 128 neurons ($\approx$ 1000 tokens). Future work should analyze how varying the patch size impacts NDT2's performance on neural activity reconstruction and behavior decoding tasks.

Table 9: *Effects of masking ratio on NDT1 performance in neural activity reconstruction.* The reported metrics quantify performance in terms of average bits per spike (bps) across all neurons from a selected session. A higher bps value indicates better performance.

| Masking | Mask Ratio = 0.1 | | | | Mask Ratio = 0.3 | | | | Mask Ratio = 0.6 | | | |
|---|---|---|---|---|---|---|---|---|---|---|---|---|
| | Co-Smooth | Forward Pred | Intra-Region | Inter-Region | Co-Smooth | Forward Pred | Intra-Region | Inter-Region | Co-Smooth | Forward Pred | Intra-Region | Inter-Region |
| Temporal (Baseline) | 0.87 | 0.73 | -0.19 | 0.44 | 1.21 | 0.88 | 0.31 | 0.52 | 0.92 | 0.88 | 0.42 | 0.56 |
| Neuron | 1.41 | -0.17 | 0.29 | 0.55 | 1.38 | -0.08 | 0.27 | 0.79 | 1.25 | 0.02 | 0.66 | 1.10 |
| Causal | 0.92 | 0.82 | 0.14 | 0.46 | 0.92 | 0.82 | 0.14 | 0.46 | 0.92 | 0.82 | 0.14 | 0.46 |
| Intra-Region | -3.79 | -0.76 | 0.96 | -4.06 | -3.62 | -0.60 | 0.86 | -3.80 | -2.18 | -0.33 | 0.43 | -2.33 |
| Inter-Region | 1.12 | 0.49 | -0.07 | 1.14 | 1.12 | 0.49 | -0.07 | 1.14 | 1.12 | 0.49 | -0.07 | 1.14 |
| MtM (Prompted) | 1.31 | 0.79 | 0.92 | 1.17 | 1.24 | 0.74 | 0.77 | 1.08 | 0.93 | 0.83 | 0.71 | 0.88 |

**Ablation study on positional embeddings.** We perform an additional experiment to evaluate the impact of the positional embedding on MtM's performance for different downstream tasks. In Table 10, we compare the effects of RoPE and learnable positional embeddings on the NDT1 architecture pretrained using the MtM method. The results demonstrate that RoPE consistently outperforms learnable positional embeddings across all downstream tasks.

Table 10: *Ablation study examining the impact of RoPE vs. learnable positional embeddings in NDT1 pretrained with the MtM method across various downstream tasks.* The displayed results are averaged over three sessions.

| | Activity Reconstruction | | | | Behavior Decoding | |
|---|---|---|---|---|---|---|
| | Co-Smooth | Forward Pred | Intra-Region | Inter-Region | Choice | Whisker Motion Energy |
| Learnable PE | 0.74 | 0.27 | 0.54 | 0.54 | 0.87 | 0.69 |
| RoPE | 0.78 | 0.36 | 0.60 | 0.60 | 0.90 | 0.70 |

**Co-smoothing evaluation metrics.** We use bits per spike as a metric to evaluate model performance across co-smoothing, forward prediction, and both inter-region and intra-region tasks. The bits per spike for a neuron is defined as follows:

$$\text{bits/spike} = \frac{1}{n_{\text{sp}}\log 2}(\mathcal{L}(r; X_{n,t}) - \mathcal{L}(\bar{r}_{n,:}; X_{n,t})),$$

where $X_{n,t}$ represents the neural activity of the target neuron $n$ at time $t$, $r$ is the inferred firing rate from the model, $\bar{r}_{n,:}$ is the mean firing rate for neuron $n$, and $n_{\text{sp}}$ is its total number of spikes. We then average the bits per spike across all neurons in a test session to get the final performance.

The bits per spike metric is closely related to the deviance, a standard metric generally used in Statistics, e.g., generalized linear models (GLMs) use deviance as a goodness-of-fit measure for a statistical model. The deviance compares the goodness-of-fit of the model of interest, e.g., MtM, to a baseline null model, where the goodness-of-fit is measured by the model log likelihood. The bps is a normalized version of the deviance metric, which compares the model predictions to the average firing rate of the neuron for the trial. The bps further normalizes the deviance by the spike count so that the metric can be comparable across neurons regardless of whether the neurons are active or inactive.

**Chance level for behavior decoding.**    For binary choice, we report the test accuracy that will be reached when constantly predicting the train set's majority class in Table 11. For whisker motion energy, we report the test $R^2$ that will be reached when consistently using the train set's trial average. Despite imbalanced datasets (e.g., c7bf2d49), Figure 5 shows that decoding accuracy and $R^2$ from selected regions are generally larger than these chance levels.

Table 11: *The chance level for decoding choice and whisker motion energy in each held-out session.*

| Chance Level | 5dcee0eb | c7bf2d49 | 3d9a098 | d57df551 | 824cf03d |
|---|---|---|---|---|---|
| Choice | 0.58 | 0.91 | 0.52 | 0.49 | 0.62 |
| Whisker Motion Energy | 0.44 | 0.30 | 0.59 | 0.43 | 0.39 |

