# OpenReview forum: "Towards a "Universal Translator" for Neural Dynamics at Single-Cell, Single-Spike Resolution"
_NeurIPS.cc/2024/Conference — NeurIPS 2024 poster_

### Official Review · Reviewer_P8rK · 2024-06-17

**Soundness:** 4
**Presentation:** 3
**Contribution:** 3
**Rating:** 6
**Confidence:** 4

**Summary:**

This work develops a multi-task-masking (MtM) approach, based on a self-supervised Transformer, that masks and reconstructs activity across different dimensions for neural spiking data learning. Evaluated on the International Brain Laboratory dataset, the model improves tasks such as single-neuron and region-level activity prediction, forward prediction, and behavior decoding. This method enhances performance and generalization, advancing towards a comprehensive foundation model for the brain.

**Strengths:**

1. The deep thinking about foundation model in neurosciece.

2. The model significantly improves generalization by modeling across different brain areas, individuals, and sessions.

3. The multi-task-masking (MtM) strategy effectively enhances the capabilities of the Transformer model.

**Weaknesses:**

1. Lack of Qualitative Analysis:

There is a lack of qualitative analysis regarding the neural dynamics obtained from population neural activity across different individuals, such as visualizing features output by NDT. Are there similar patterns? Are there similarities between single-neuron and population-level activities?

2. Insufficient Ablation Studies:

a) What are the differences between training on multi-region data versus single-region data? When training with data from three brain areas, is there a phenomenon where the activity of one brain region dominates?

b) What are the performance differences when using multi-session data from a single individual versus cross-individual multi-session data?

**Questions:**

1. Typically, cross-individual modeling aims to explore common patterns among different individuals, while cross-area modeling aims to understand information interactions between different brain areas. How much do you think the current foundation model can contribute to these explorations?

2. Generally, in neuroscience, there is a greater interest in understanding the interactions with other internal brain regions, which would be helpful for functional modeling in network level. What insights can the current foundation model provide for research in these areas?

**Limitations:**

see Weaknesses.

---

> ### Author Rebuttal · Authors · 2024-08-07
>
> **Weaknesses:**
> 1. We agree with the reviewer that our submission was lacking in qualitative analysis. To partially address this, we provide visualizations of the rastermap reconstructions for all predictive tasks in Figure 1 of our one page pdf. For the final version of our paper, we will include rastermap visualizations for many different animals/sessions, so readers can see differences in neural activity across different individuals. We believe there might be some confusion about the neural dynamics learned in our models. Since NDT1 is a transformer-based model, neural dynamics are learned implicitly. We are not explicitly learning neural dynamics like a state-space model would. So it is not possible to visualize the dynamics directly in a transformer. Building transformer architectures with explicit latent dynamics is an interesting future direction.
>
> 2. These are very interesting questions! (a) When training on single region data, we would recommend just using the neuron and causal masking, as region-level predictions are not informative in this setting. When training on data from multiple brain areas, we uniformly sample brain areas for inter/intra region masking and then we average the loss per-neuron and per-timestep; this makes sure that no brain region dominates the loss. (b) We are using the IBL repeated sites dataset [1] which only have repeated sessions for one individual so we cannot easily address the difference in performance when using multi-session data from a single individual versus cross-individual multi-session data. Using the full public IBL brain-wide map dataset [2] would allow us to test this phenomena and is an exciting future direction.
>
> [1] International Brain Laboratory, et al. "Reproducibility of in-vivo electrophysiological measurements in mice." bioRxiv (2022): 2022-05.
>
> [2] International Brain Laboratory, et al. "A brain-wide map of neural activity during complex behaviour." Biorxiv (2023): 2023-07.
>
>
> **Questions:**
> 1. This is an interesting question and something we have been thinking about as well. As a first attempt to answer this question, we ran a new experiment where we try to predict region A from region B for all regions in a test session using a model trained with MtM. As this is different from our current inter-region predictive task (which predicts one region from all other regions), we fine-tuned our pretrained MtM model with a new inter-region task where we randomly dropout some of the regions when predicting one region. The prediction matrix (region A to region B for all regions) is shown in Figure 2 of the one page PDF. It can be seen that there is interesting structure in the matrix where some regions can predict each other well, for example in visual areas. We believe that exploring the "functional connectivity" of different areas using MtM is an interesting future work.
>
> 2. Good question! Models trained with MtM will be immediately useful for understanding interactions between different brain areas. As discussed above, we were able to construct a "functional connectivity" matrix using MtM that can show how predictive different regions are from each other. We believe it may also be possible to examine the attention weights when predicting region A from region B to interpret temporal delays between the two areas.
> Using MtM to explore common patterns among different individuals is more challenging. An interesting result of our paper is that when pretraining with more individuals, the single individual performance improves after fine-tuning. This indicates that there are common patterns among individuals that are being captured by MtM. If we were to utilize the MtM objective with a latent variable model like LFADS, it may be possible to look at how dynamics in different areas differ among different individuals. This is an exciting figure direction we hope to explore.

---

> > ### Comment · Reviewer_P8rK · 2024-08-08
> >
> > The rebuttal PDF meets my expectations. Although the learning of neural dynamics is implicit, some patterns appear to be discernible. Please consider further joint analysis with interpretable methods in the future work, otherwise this kind of work is useless for neuroscience. I would like to maintain my score. Good Luck!

---

> > > ### Author Response · Authors · 2024-08-08
> > > **Applying MtM to interpretable models of neural activity (in Global Response)**
> > >
> > > We thank the reviewer for responding back to us so quickly. We believe that MtM can easily be combined with more interpretable models of neural populations. For example, as part of our Global Response, we combined our MtM learning objective with LFADS, an interpretable, explicit dynamics model. We were able to improve the performance of LFADS on all predictive tasks introduced in the paper. Although we have no more space for figures, interpreting the latent factors learned by LFADS when trained with MtM is an exciting future direction and something we can do in the updated manuscript. We hope the ability of MtM to be applied to interpretable latent factor models like LFADS can improve the reviewer's score of our manuscript.
> > >
> > > Although explicit dynamics models are very important for neuroscience, we still believe that purely predictive models like transformers are also useful. Neural decoding/encoding algorithms have been developed for many years to determine how much information about a stimulus or behavior is encoded in a neural population [1]. Much like these algorithms, we are interested in determining how much information each brain area has about other brain areas. Understanding which brain areas predict each other best can allow us to make hypotheses about distributed neural computation.
> > >
> > > [1]​​ Paninski, Liam, Jonathan Pillow, and Jeremy Lewi. "Statistical models for neural encoding, decoding, and optimal stimulus design." Progress in brain research 165 (2007): 493-507.

---

### Official Review · Reviewer_xe9o · 2024-07-08

**Soundness:** 3
**Presentation:** 3
**Contribution:** 2
**Rating:** 6
**Confidence:** 4

**Summary:**

This work proposes a transformer architecture (based on previously existing ones) and, more interestingly, a training procedure that, when applied to spiking neural data, should result in a foundation model for spiking neural data. The clever bit about the procedure is that the learning model is asked to reconstruct differently-masked portions of the data: activity of one held-out neuron, future activity of a neuron, activity of a neuron based on neurons in the same region, activity of a neuron based on activity of neurons in other regions. During training these tasks are presented to the model interchangeably, so it has to learn to do all of them (although with a help of an embedding vector that can decode for the type of the task).

The results show that the proposed model achieves better numbers in a variety of tests, however I found it hard to evaluate the significance of the result using the reported metric of bits per spike and how it translates to real-world performance. Please see my comments below and I hope to learn more about this from authors' rebuttal.

**Strengths:**

* In my opinion, this work is a step in a very important direction, indeed with the wealth of neural data being collected foundation models for various modalities can become very important tools for modelling brain dynamics

* Clearly and simply written, easy to read and follow

**Weaknesses:**

* I would like to see more about the architecture of the transformer (even though it is based on existing model), ideally a more detailed visualisation, this would help to imagine how exactly the data is going in, how the masking is applied, etc.

* I am not too happy about the bit-per-spike metric being the only one reported. It is fine to have one number to be able to characterize and compare the performance succinctly, but it's awkward to translate back to "how well does it work, really?". I think it would be great if the paper would include actual raster plots of good / average / worst reconstructions of the masked neuron / future / intra-region / inter-region regions -- this would allow a neuroscientist to decide with a quick glance whether "wow I should start using this" or "nah I am gonna wait till NeurIPS 2025"

**Questions:**

184: Please include a formal definition of "bits per spike"

Table 1: How a "bits per spike" measure can be negative? What does it mean?

Figure 5: Please explain "Choice" and "WME" and axis units in the caption and/or on the figure. Perhaps it would be better to fix the range on the axis, so that one could easily estimate also the variability between the sessions and MtM improvement in the sessions.

**Limitations:**

The authors have adequately highlighted some limitations, but I would like to hear a bit more about the limitations of the reconstructed signal from neuroscientific perspective: are they usable? Do they brake or maintain some important characteristics of spiking signal?

---

> ### Author Rebuttal · Authors · 2024-08-07
>
> **Weaknesses:**
>
> 1. We agree with the reviewer that this is a important detail missing from our paper. To address this, we added schematics for NDT1 and NDT2 that demonstrate how tokenization and neural reconstruction works (please see Figure 3 in the 1 page PDF). We will add this figure to the supplement in the final version of the paper.
>
> 2. We agree with the reviewer that the lack of visualizations is a problem with our original submission. To partially address this, we provide visualizations of the rastermap reconstructions for all predictive tasks in Figure 1 of our one page pdf. The difference in the bps metric for temporal masking and MtM can clearly be seen in the region-level predictive tasks and can also be seen in neuron and forward prediction tasks. We will make sure to add rastermap visualizations for many different sessions in the supplement of our final paper. Hopefully, this will convince readers to try out our approach for NeurIPS 2024! :-)
>
> **Questions:**
> 1. We define the bits per spike for a neuron with the same equation as [1] on the bottom of page 6. We then average the bits per spike for each neuron in a test session to get the final performance. We will include a version of this equation in the updated manuscript.
>
> 2. The bits per spike (bps) can be negative when the predicted rates for a neuron are less informative than just predicting the average firing rate of the neuron for the trial. This can be seen in the above equation where we subtract the likelihood of the neural activity assuming an average firing rate from the likelihood of the neural activity with the predicted rates of the model. The bps also tends to be more negative when trying to predict low-firing rate neurons.
>
> 3. "Choice" refers to the choice decoding accuracy for each method. "WME" refers to the whisker motion energy decoding $R^2$ for each method . We will improve the axis labels in the final version of the paper.  We will also try fixing the axis for each session as suggested by the reviewer.
>
> [1] Pei, Felix, et al. "Neural latents benchmark'21: evaluating latent variable models of neural population activity." arXiv preprint arXiv:2109.04463 (2021).

---

> ### Comment · Reviewer_xe9o · 2024-08-08
>
> Figures 1 and 3 in the rebuttal PDF are very nice and informative. Thank you for providing those.
> I think Figure 1 should even be part of the manuscript itself (with more visualizations in the supplementary materials)
>
> Q1: It seems to me that MtM smoothes adjacent time bins quite a lot. Ground truth often has almost empty time bins, followed by a sharp increase in the firing rate right in the next bin. But in MtM's reconstruction the change of firing rate is always gradual from bin to bin. And that worries me because sharp increases and sudden spikes or drops are very important for actual neuroscientific insight. Why dies this smoothing occur? Is adjacency in the time series plays some sort of a special role in a transformer? Or is it due to the loss function? Can you define a loss function that would strive to preserve sharp changes?
>
> Q2: I am still somewhat unhappy with the "bits per spike" measure :), however seeing the actual definition was very helpful and for sure needs to be part of the manuscript. The issue I take with it - what is the benefit of adding this additional layer of abstraction and measure via Poisson model? Why not go super simple and just count the number of spikes the prediction got wrong? (and then average in a suitable manner to make comparable across). Spikes are binary, and this makes it easy to count them. Maybe it's just me, but when I see a metric like "bps: 0.51" I have no idea how good or bad is it. Maybe if you provide info on what the bps would be for a perfect reconstruction? I guess what I am looking for is a more intuitive number here, like "73% of spikes are where they should be, 27% are missing / extra" or "the number of spikes MtM predicts within each bin is off by 2.7 spikes per bin, or 14%".
>
> Basically I am still not able to understand when I am looking at "Baseline bps: 0.03" vs. "MtM bps: 0.51" should I be impressed? How impressed should I be? Reconstruction-wise both Baseline and MtM look pretty far away from the ground truth.
>
> Q3 (exploratory, just for fun): Do I understand correctly that currently tokens are based on firing rates? Is yes, then my question is would it be possible to  use analgues of "spiking words" or "spiking syllables" of fixed length instead? Something like
> ```
> ||||__|||| -> token 1
> |||||||___ -> token 2
> ___||||___ -> token 3
> ... etc
> ```
> Where "|" are spikes and "_" are non-spikes. To basically build a vocabulary of all possible 10ms spiking patterns and use those as tokens?
>
> Or, actually, why use bins at all? What would happen if you use spiking sequences as-is, so that the time step is small enough to just be either | or _ ?

---

> > ### Author Response · Authors · 2024-08-08
> > **Response to additional questions from Reviewer xe9o - Part 1**
> >
> > We thank the reviewer for getting back to us so promptly! We agree that Figure 1 should be a part of the manuscript and we will include a version of this figure in the main text of the updated manuscript. We believe there might be a misunderstanding about the goals of statistical models of neural data that we hope to clarify in the following response. Our response will be in two parts due to character constraints (we will answer Q2 and Q1 in the first comment). If we answer all questions satisfactorily, we hope the reviewer would consider raising their score.
> >
> > **Q2**
> >
> > We are glad the reviewer found our definition of bps helpful and we will make sure to add this in the updated manuscript. We suspect there might be a misunderstanding here about prior literature on neural population models. As described in [1], there is a rich literature of population models that utilize either Gaussian or, more recently, Poisson observation models (please see Table 2 of [1]) to predict neural spiking activity. The goal of neural population models is to predict the underlying firing rates of each neuron because the exact spike times are believed to be observations from a stochastic process. Similar to this seminal paper [2], we choose to model the observed spikes as samples from an inhomogeneous Poisson process whose rate corresponds to the inferred firing rate for the given neuron. To measure the quality of fit of our Poisson process, it is most natural to use a version of the log-likelihood such as the bps metric utilized in the Neural Latents Benchmark [3]. Using this commonly agreed upon metric makes sure that our work is consistent and comparable with existing research on neural population models.
> >
> > In addition, the "bits per spike" metric is closely related to the "deviance", a standard metric generally used in Statistics, e.g., generalized linear models (GLMs) use deviance as a goodness-of-fit measure for a statistical model. The deviance compares the goodness-of-fit of the model of interest, e.g., MtM, to a baseline null model, where the goodness-of-fit is measured by the model log likelihood. The bps is simply a normalized version of the deviance metric, which compares the model predictions to the average firing rate of the neuron for the trial. The bps further normalizes the deviance by the spike count so that the metric can be comparable across neurons regardless of whether the neurons are active or inactive.
> >
> > Along with the bps, we would be happy to report the R-squared in the revision to provide further intuition, but note that the R-squared depends strongly on bin size (as bin size gets small so will R-squared, since the precise timing of spikes is likely not predictable in this setting).
> >
> > **Q1**
> >
> > In Figure 1 of the one page PDF, the ground-truth rastermaps are the spike counts of the trial while the MtM/NDT rastermaps are the inferred firing rates. These quantities are not directly comparable because we expect the underlying firing rate of a neuron to vary smoothly while its observed spikes (which are assumed to be stochastic) may have sharp increases. A great example of this phenomena is shown in [4] where they demonstrate that when the inferred firing rates exactly fit each spike time, this is actually overfitting to the neural data. They introduce a novel dropout strategy, similar to MtM's neuron masking, that reduces this overfitting.
> > In fact, it has been shown by [2] (in Figure 2e) that the smoothed firing rates from a Poisson model actually allow for decoding animal behavior more accurately than the raw spike counts. Whether to smooth or not involves a bias-variance tradeoff. More smoothing means more bias and less variance. When we optimize for prediction accuracy, we are implicitly optimizing this tradeoff, and empirically we (and others) find that some smoothing (bias) leads to better predictions.
> >
> > [1] Hurwitz, Cole, et al. "Building population models for large-scale neural recordings: Opportunities and pitfalls." Current opinion in neurobiology, 2021.
> >
> > [2] Pandarinath, Chethan, et al. "Inferring single-trial neural population dynamics using sequential auto-encoders." Nature methods, 2018.
> >
> > [3] Pei, Felix, et al. "Neural latents benchmark'21: evaluating latent variable models of neural population activity." arXiv 2021.
> >
> > [4] Keshtkaran, Mohammad Reza, et al. "A large-scale neural network training framework for generalized estimation of single-trial population dynamics." Nature Methods, 2022.

---

> > > ### Author Response · Authors · 2024-08-08
> > > **Response to additional questions from Reviewer xe9o - Part 2**
> > >
> > > **Q3**
> > >
> > >
> > > Before answer the question, we want to clarify that MtM is agnostic to the underlying transformer architecture and that the tokenization scheme will depend on the architecture that is used.
> > >
> > > To answer the question, a token for NDT is the observed spike counts for a single 20ms timestep of the population (please see Figure 3 in [5]). This means that NDT is using the exact spike times (up to 20 ms resolution) to fit the model. The reviewer's suggested tokenization scheme would be comparable to NDT with more fine binning (1 ms bins). While this is feasible, given the sparsity of neural data, this would give rise to many more empty bins and would also be computationally prohibitive (i.e., self-attention scales quadratically with the number of tokens). As the reviewer mentioned, "why use bins at all?". Treating each spike as a token, as done in the POYO paper [6], is also a potential solution. As this is computationally prohibitive, they adopted a perceiver IO architecture [7] so the attention is only applied to a set of latent tokens, which is much lower dimensionality. While this architecture is a promising step in the right direction, it has only ever been used as a supervised model to decode behavior from spikes. Adapting POYO and more interesting transformer architectures to be self-supervised such that they can be used with MtM is an exciting future direction.
> > >
> > >
> > > [5] Ye, Joel, and Chethan Pandarinath. "Representation learning for neural population activity with Neural Data Transformers." arXiv, 2021.
> > >
> > > [6] Azabou, Mehdi, et al. "A unified, scalable framework for neural population decoding." NeurIPS 2024.
> > >
> > > [7] Jaegle, Andrew, et al. "Perceiver io: A general architecture for structured inputs & outputs." arXiv, (2021).

---

> > > > ### Comment · Reviewer_xe9o · 2024-08-10
> > > >
> > > > Thank you for explanations and clarifications. They were helpful to alleviate some of my concerns and I will raise my overall score a bit. Good luck!

---

### Official Review · Reviewer_JcgB · 2024-07-13

**Soundness:** 3
**Presentation:** 4
**Contribution:** 3
**Rating:** 6
**Confidence:** 4

**Summary:**

The authors propose a self-supervised approach to building a foundation model of single-trial neural population dynamics. The approach utilizes existing transformer architectures, which are trained using “multi-task-masking” (MTM), which alternates between several scales of prediction tasks, including co-smoothing, causal prediction, inter-region prediction, and intra-region prediction. The approach is demonstrated on the IBL repeated site dataset, where activity prediction, forward prediction, and behavior decoding using a fixed architecture are improved using MTM training relative to a baseline where training only utilizes temporal masking.

**Strengths:**

The work is original and forward thinking. The writing is admirably clear and well referenced.

**Weaknesses:**

The paper would benefit from:
- Comparing to a wider range of models of neural population dynamics (e.g., LFADS, GPFA, mDLAG, etc). Many previous approaches have reported co-smoothing performance (e.g., Neural Latents Benchmark). How does MTM-NDT1 compare against those previous approaches on co-smoothing? How does the proposed approach compare to multi-area models (e.g., mDLAG) for on the inter-region co-smoothing task?
- More fully describing the details of the “temporal masking” baseline. The details are referenced out to a citation rather than described within the text. Since this baseline appears throughout all of the results, it seems important to describe it within this paper.

**Questions:**

- What is the motivation for building a foundation model for neural data? What are potential applications for such a model? Can such a model teach us anything about the brain?

**Limitations:**

The authors adequately state the limitations of their modeling approach, training paradigm, and datasets.

---

> ### Author Rebuttal · Authors · 2024-08-07
>
> **Weaknesses:**
> 1. We thank the reviewer for this suggestion to compare to a wider range of neural population models. We want to clarify that MtM can be used as a learning objective for any architecture (not just transformers). To demonstrate this, we ran a new experiment using the LFADS architecture (a sequential VAE) trained with both temporal masking and MtM. The results can be seen in Table 1 of the Global Response. MtM training improves all activity prediction metrics for LFADS especially intra-region masking and inter-region masking. The best performing model is still NDT1-MtM, but the LFADS-MtM model is quite competitive. This experiment provides evidence that MtM is a model-agnostic objective that can teach region-level population structure to both transformer and RNN architectures. We also attempted to compare MtM to mDLAG on three regions (VISa2/3, VISa4, VISa5) of a single test session. For these regions, there were a total of 42 neurons, 828 trials, and 100 time bins (20 ms bin size). We trained mDLAG for 5.5 hours (5000 EM iterations) and the model did not converge. A visual inspection shows that the learned latent factors for each region were nearly identical and that activity reconstruction from the learned latents consistently led to low co-bps (negative to zero). This experiment provides evidence that scaling mDLAG to full IBL datasets with tens of regions and thousands of neurons is not feasible. We will provide all details for this experiment in the updated manuscript.
>
>
> 2. We agree with the reviewer that this is an important detail missing from the initial submission. We will make sure to add a discussion of both temporal masking for NDT1 and random token masking for NDT2 in the final submission. We also plan to add schematics for the NDT1 and NDT2 architectures in the updated manuscript (Figure 3 of the one page PDF) to demonstrate how they tokenize and reconstruct neural data.
>
> **Questions:**
> 1. This is a great question and something we are also thinking hard about! We believe that by incorporating data from many different sources, a neural foundation model can help us answer important questions about cross-individual (and even cross-species) neural variability. We also believe that neural foundation models, which are trained on many different combinations of brain regions, can eventually be used to predict missing brain regions in new recordings. We also believe that by training neural foundation models on healthy animals, we can test hypotheses about neural activity in diseased animals. This advancement could potentially lead to new treatments and interventions for neurological disorders. Finally, it will be possible to fine-tune pretrained foundation models on new tasks (brain computer interfaces, etc.) to achieve state-of-the-art performance [1]. We believe our work is a step towards realizing this vision and we are hopeful that the community can start working with and building similar models. We will improve our discussion of foundation models of neural data in the updated manuscript.
>
> [1] Azabou, Mehdi, et al. "A unified, scalable framework for neural population decoding." Advances in Neural Information Processing Systems 36 (2024).

---

> > ### Comment · Reviewer_JcgB · 2024-08-12
> >
> > Thank you for the thoughtful rebuttal. I believe the final product will indeed be improved by incorporating the ideas from these reviews and rebuttals. I stand by my score.

---

> > > ### Author Response · Authors · 2024-08-12
> > > **Response to Reviewer JcgB**
> > >
> > > We thank the reviewer for their positive response to our rebuttal. If there is anything we can further clarify or do to improve the reviewer's overall assessment of our work, please let us know.

---

### Official Review · Reviewer_BBJQ · 2024-07-14

**Soundness:** 3
**Presentation:** 3
**Contribution:** 2
**Rating:** 6
**Confidence:** 4

**Summary:**

The paper presents a large-scale model pretrained on the International Brain Laboratory (IBL) corpus containing multi-region, multi-animal spiking activity of mice during a decision-making task. It introduces a self-supervised learning approach with a novel masking scheme called MtM which alternates between masking and reconstructing activity in timesteps, neurons and brain regions. The masking strategy was shown to be helpful in learning different structures in neural population activity and outperforms temporal masking scheme of previous models. The paper shows positive scaling results on held-out animals as the number of training animals increases.

**Strengths:**

1.	The paper tackles an important problem of building “foundation models” for neuroscience at the level of spiking activity that can help with the understanding of the brain structure and facilitate the decoding of animal behaviors.
2.	The proposed MtM method is novel and demonstrated effectiveness over the existing temporal masking method in learning different structures of neural activity.
3.	Thorough evaluation on multi-region, multi-subject IBL dataset provides valuable insights for the neuroscience community.
4.	The manuscript is well-written. Text and figures are logical and easy to follow.

**Weaknesses:**

1.	The number of baselines is quite limited. Only two transformer architectures (NDT1 and NDT2) were benchmarked, and the proposed MtM method is mainly compared with one baseline of temporal masking. It appears that MtM could be used as the training objective for similar existing models that also assume a Poisson emission model, e.g. RNN-based LFADS (Pandarinath et al. 2018) and other models in the NLB benchmark (Pei et al. 2021). Evaluating MtM on more architectures could be helpful to gauge if MtM effectiveness is truly model-agnostic as the authors noted or is biased more toward certain type of architecture.
2.	The dataset only consists of one behavior task (decision making), while existing pretrained models POYO and NDT2 referenced in the paper were trained/evaluated on multiple behavior (motor) tasks, even across species (monkey vs. human). It is unclear if and how the community could benefit from finetuning the proposed pretrained model on other datasets with a different behavior paradigm.
3.	Some experiment details are missing or not presented clearly (elaborated in Questions section below).

**Questions:**

1.	It would be helpful to compare the proposed model with more NLB baselines to justify the architecture choice. Does NDT1 with MtM significantly outperform simpler baselines like RNN with MtM to justify the computation cost? Even knowing how well baselines like MLP at directly decoding the behavior could help readers better evaluate the strength of the proposed model.
2.	It is not clear if the pretrained model would provide some leverage on other datasets with different behavior structures. Would it outperform a model trained from scratch on the new dataset?
3.	It might be worth making it clear that “task” in “multi-task” used in the paper refers to the machine predictive task (decoding choice and motion energy), rather than animal behavior task (decision making – the only behavior investigated).
4.	Equation (1): Does the transformer use any kind of positional encoding? If so, what type of positional encoding?
5.	For finetuning on a held-out session, I assume the reported metrics were calculated on the test trials within the session. How many trials are there in each train/val/test split of each session?
6.	Line 174: possibly a typo, should be 1% of the trial (20ms).
7.	How was choice decoded from $\hat{X}$? I assume from Figure 1 that $\hat{X}$ is a N x T matrix, where N is number of neurons and T is number of timestep. If so, at which timestep of $\hat{X}$ that the linear layer was applied to decode the choice?
8.	Similarly, was the motion energy decoded at each timestep of $\hat{X}$ or at some particular point during the trial?
9.	Figure 5: what is the chance level for each session? For imbalance datasets, the chance level might not be 0.5, especially if the data was preprocessed for quality control in some ways. If so, it’s worth plotting the chance level in the figure.

**Limitations:**

The authors have addressed the limitations in the manuscript.

---

> ### Author Rebuttal · Authors · 2024-08-07
>
> **Weaknesses:**
> 1. We thank the reviewer for the suggestion to run more baseline architectures with the MtM objective. Our goal was to compare masked modeling approaches for neural population modeling which just includes temporal masking as a baseline. Based on this feedback, however, we ran a new experiment where we trained the LFADS architecture (a sequential VAE) with both temporal masking and MtM. The results can be seen in Table 1 of the Global Response. MtM training improves all activity prediction metrics for LFADS especially intra-region masking and inter-region masking. The best performing model is still NDT1-MtM, but the LFADS-MtM model is quite competitive. This experiment provides evidence that MtM is a model-agnostic objective that can teach region-level population structure to both transformer and RNN architectures.
> 2. We agree with the reviewer that there is a lack of task and species diversity in our original submission. To partially address these concerns, we fine-tuned our 34-session pretrained NDT1-MtM on the Neural Latents Benchmark (NLB) MC_RTT dataset. For details on this experiment, please see the Global Response. With a single private submission to the leader board, the MtM performance (.19 bps) significantly improves upon NDT's previous best performance (.16 bps). These results show promising transfer of our pretrained MtM models to a new task and species. We also want to point out that while the IBL datasets lack species and task diversity, they have high diversity of recorded brain areas. Brain region diversity is an unexplored axis for large-scale pretraining as the POYO and NDT2 papers only train on data from motor areas. We wanted to tackle this challenge first before moving on to cross-species transfer. However, we believe that these approaches are complementary and that building large-scale models that generalize across different tasks, species, and brain areas will be an exciting future direction.
>
> **Questions:**
> 1. We agree with the reviewer that our paper lacks a simpler NLB baseline and have implemented an LFADS baseline model trained with temporal and MtM masking (results in Table 1 of the Global Response). NDT1 is still the top performing model when trained with the MtM objective although LFADS-MtM is competitive. While we agree that the LFADS RNN architecture is simpler, transformers have been shown to scale extremely well to massive datasets and also have benefits for computational efficiency (please see Figure 6 of [1]). We will clarify these reasons for using transformers in the updated manuscript. We did not include an additional MLP baseline for behavior decoding because we are decoding from the inferred rates and, therefore, would not expect to do better than an MLP. This is the strategy used in the Neural Latents Benchmark [2] to evaluate behavior prediction.
>
> 2. This is a great question and something we are interested in exploring as future research. To attempt to answer this question, we fine-tuned our 34-session pretrained NDT1-MtM on the Neural Latents Benchmark (NLB) MC_RTT dataset. All details for this experiment can be found in the Global Response.
>
> 3. Yes, there is some overloading of terminology in our paper between behavioral "tasks" in neuroscience and the predictive "tasks" of MtM. In MtM, there are multiple neuron-level and region- level predictive tasks. We will make this distinction clear in the final version of the paper.
>
> 4. Yes, both NDT1 and NDT2 use a positional embedding layer for each token. NDT1 uses the timestep for the positional encoding. NDT2 uses 2 types of position encoding: (1) timesteps and (2) neuron group identity. We did not include the positional embedding step in Equation (1), but we can correct this omission in the final version. We also created schematic diagrams for both NDT1 and NDT2 (see Figure 3 in the one page PDF) which we will add to the supplement to improve our description of the underlying architectures.
>
> 5. In the held-out sessions, there are 397 ~ 579 trials in the train split, 57 ~ 83 trials in the validation split, and 114 ~ 166 trials in the test split.
>
> 6. Yes, good catch. This should read "10% of the trial (200ms)". We will correct this in the final version.
>
> 7. For each trial, we predict the binary choice using the entire $N \times T$ matrix $X$, which represents the spiking activity of all neurons across all timesteps.
>
> 8. In each trial, we use the entire $N \times T$ matrix $X$ to predict the motion energy at every timestep $t = 1, 2, …, T$, where $T$ is the total number of timesteps.
>
> 9. For binary choice, we report the test accuracy that will be reached when constantly predicting the train set's majority class. For whisker motion energy, we report the test $R^2$ that will be reached when consistently using the train set’s trial average. Despite imbalanced datasets (e.g., c7bf2d49), Figure 5 shows that decoding accuracy and R2 from selected regions are generally larger than these chance levels.
>
>     |      Chance Level     | 5dcee0eb | c7bf2d49 | 3d9a098 | d57df551 | 824cf03d |
>     |:---------------------:|:--------:|:--------:|:-------:|:--------:|:--------:|
>     |         Choice        |   0.58   |   0.91   |   0.52  |   0.49   |   0.62   |
>     | Whisker Motion Energy |   0.44   |   0.30   |   0.59  |   0.43   |   0.39   |
>     |                       |          |          |         |          |          |
>
> [1] Ye, Joel, and Chethan Pandarinath. "Representation learning for neural population activity with Neural Data Transformers." arXiv, 2021.
>
> [2] Pei, Felix, et al. "Neural latents benchmark'21: evaluating latent variable models of neural population activity." arXiv preprint arXiv:2109.04463 (2021).

---

> > ### Author Response · Authors · 2024-08-13
> > **Response to Reviewer BBJQ**
> >
> > We hope the reviewer found our rebuttal helpful. If there is anything we can further clarify, we are happy to do so!

---

> > > ### Comment · Reviewer_BBJQ · 2024-08-14
> > >
> > > Thank you for the explanation. The extra experiments alleviated my concerns. I've increased my rating accordingly.

---

### Author Rebuttal · Authors · 2024-08-07

We thank the reviewers for the thoughtful and detailed feedback on our manuscript. We are excited to hear that the reviewers thought that our work "is original and forward thinking" (**JcgB**), represents "deep thinking about foundation models in neuroscience" (**P8rK**), and marks a "step in a very important direction" (**xe9o**). They remarked that "The proposed MtM method is novel" (**BBJQ**), and we provided a "thorough evaluation on the multi-region, multi-subject IBL dataset that provides valuable insights" (**BBJQ**). Multiple reviewers also comment on the clarity of the writing and that "The writing is admirably clear and well referenced." (**JcgB**)

We have done our best in the rebuttal to address any concerns. To this end, we ran four new experiments and added additional figures (in the one page pdf) detailed below:

* An additional population model baseline: LFADS
  * We have added an experiment where we trained the LFADS architecture (a sequential VAE) on 5 test sessions using the temporal masking baseline and our MtM learning objective. The results of the LFADS analysis are shown in Table 1 of our global response. We found that the LFADS architecture trained with MtM outperforms temporal masking on all activity prediction tasks and is competitive with NDT1 trained with MtM (although slightly worse). Similar to our original manuscript, behavior prediction is comparable between MtM and temporal masking (see Section 6.1 and Figure 5 of our manuscript for a more detailed discussion about behavior decoding). This experiment suggests that MtM is an architecture-agnostic learning objective that can be used with both transformers and RNNs. We do not plan to include any other population model baselines as we are focused on models that can do multi-session training such as NDT1, NDT2, and LFADS. We will add details about this LFADS experiment in the updated manuscript.

    | Table 1 | Co-Smooth | Forward Prediction | Inter-Region | Intra-Region | Choice | Whisker Motion Energy |
    | -------- | ------- | ------- | ------- | ------- | ------- | ------- |
    | NDT1 Baseline  |  0.55 | 0.17 | 0.18 | -0.49 | 0.87 | 0.65 |
    | NDT1 MtM| 1.00 | 0.28 | 0.83 | 0.70 | 0.85 | 0.67 |
    | LFADS Baseline | 0.86 | 0.09 | 0.50 | -0.34 | 0.86 | 0.67 |
    | LFADS MtM    | 0.87 | 0.26 | 0.76 | 0.65 | 0.82 | 0.67 |

* Visualizations of the predictions for MtM and temporal masking
  * We have added visualizations of the rastermaps for NDT1 trained with MtM and temporal masking for each predictive task: neuron, causal, inter-region, and intra-region prediction. As can be seen in Figure 1 of our one page PDF, MtM leads to better predicted rastermaps for inter/intra/neuron masking. For forward prediction, the improvement is more subtle, largely because temporal masking performs well at temporal predictive tasks. Overall, these visualizations give us confidence that MtM is a significant improvement over the baseline. We will include rastermaps for additional test sessions in the updated manuscript.

* Fine-tuning our pretrained MtM model on a monkey recording
  * While the IBL repeated site dataset used for training MtM has high diversity of brain regions, Reviewer BBJQ suggested that it lacks task/species diversity. To demonstrate that MtM can be a useful pretraining strategy for new tasks and species, we fine-tuned our 34-session pretrained NDT1-MtM model on the Neural Latents Benchmark (NLB) MC_RTT dataset. This dataset consists of spiking activity recorded from the primary motor cortex of a monkey performing a self-paced reaching task. We divided this dataset into two "brain regions" based on the NLB heldin and heldout neuron split. We then fine-tuned our model with using the MtM learning objective. With a single private submission to the leader board, the MtM performance (.19 bps) significantly improved upon NDT's previous best performance (.16 bps). We will include all details in the updated manuscript.

* Attempting to run mDLAG on a subset of IBL data
  * To address Reviewer JcgB's suggestion that we test MtM again mDLAG, a multi-region latent variable model, we attempted to run mDLAG on three regions (VISa2/3, VISa4, VISa5) of a single test session. For these regions, there were a total of 42 neurons, 828 trials, and 100 time bins (20 ms bin size). We trained mDLAG for 5.5 hours (5000 EM iterations) and the model did not converge. A visual inspection shows that the learned latent factors for each region were nearly identical and that activity reconstruction from the learned latents consistently led to low co-bps (negative to zero). This experiment provides evidence that scaling mDLAG to full IBL datasets with tens of regions and thousands of neurons is not feasible. We will provide all details for this experiment in the updated manuscript.

* We utilized MtM to estimate a "functional connectivity" matrix for three test sessions.
  * Inspired by Question 1 of Reviewer P8rK, we utilized our pretrained MtM model to ask how well individual brain areas predict each other. We fine-tuned our pretrained MtM model on each test session with a new inter-region task where we randomly dropout some of the regions when predicting one region. Then, we used MtM to predict region A from region B for all regions in the session. The prediction matrix is shown in Figure 2 of the one page PDF. Even without explicit training to predict one region from another region, there is interesting structure in the matrix  as some regions predict each other well, for example in visual areas. This experiment suggests that MtM can be used to explore the "functional connectivity" of different areas. This is an exciting future direction that we would like to explore.
* We created a schematic for the NDT1 and NDT2 architectures to clarify how these architectures tokenize and reconstruct neural data. This can be seen in Figure 3 of our one page PDF. We plan to add a version of this figure to the updated manuscript.

---

### Decision · Program_Chairs · 2024-09-25

**Decision:**

Accept (poster)

**Comment:**

This is a solid paper building towards a foundation model for neural recordings at high resolution (single spikes of single cells). Trained on neuropixel data from many animals and sessions, it enables the prediction of neural activity and behavior, and the generalization to new animals. The reviewers asked for many additional analyses and clarifications that the authors provided and that should be included in the final manuscript.